# COSIPY v1.3 - An open-source coupled snowpack and ice surface energy and mass balance model

Tobias Sauter[1], Anselm Arndt[2], and Christoph Schneider[2]

[1]    Department of Geography, Friedrich-Alexander-Universität Erlangen-Nürnberg, Wetterkreuz 15, 91058 Erlangen, Germany

[2]    Geography Department, Humboldt-Universität zu Berlin, Unter den Linden 6, 10099 Berlin, Germany

**Correspondence:** Tobias Sauter (tobias.sauter@fau.de)

   **Abstract.** Glacier changes are a vivid example of how environmental systems react to a changing climate. Distributed surface mass balance models, which translate the meteorological conditions on glaciers into local melting rates help to attribute and detect glacier mass and volume responses to changes in the climate drivers. A well calibrated model is a suitable test-bed for sensitivity, detection and attribution analyses for many scientific applications and often serves as a tool for quantifying the inherent uncertainties. Here we present the open-source coupled snowpack and ice surface energy and mass balance model in Python COSIPY, which provides a flexible and user-friendly framework for modelling distributed snow and glacier mass changes. The model has a modular structure so that the exchange of routines or parameterizations of physical processes is possible with little effort for the user. The framework consists of a computational kernel, which forms the runtime environment and takes care of the initialization, the input-output routines, the parallelization as well as the grid and data structures. This structure offers maximum flexibility without having to worry about the internal numerical flow. The adaptive sub-surface scheme allows an efficient and fast calculation of the otherwise computationally demanding fundamental equations. The surface energy-balance scheme uses established standard parameterizations for radiation as well as for the energy exchange between atmosphere and surface. The schemes are coupled by solving both surface energy balance and subsurface fluxes iteratively such that consistent surface skin temperature is returned at the interface. COSIPY uses a one-dimensional approach limited to the vertical fluxes of energy and matter but neglects any lateral processes. Accordingly, the model can be easily set up in parallel computational environments for calculating both energy balance and climatic surface mass balance of glacier surfaces based on flexible horizontal grids and with varying temporal resolution. The model is made available on a freely accessible site and can be used for non-profit purposes. Scientists are encouraged to actively participate in the extension and improvement of the model code.

# 1  Introduction

Glacier variations are of great interest and relevance in many scientific issues and application such as climate sciences, water resources management and tourism. In order to identify the climatic drivers for past, current and future changes, process understanding, observations and models of glacier mass change need to be combined appropriately. Schemes that relate the surface mass balance of snow and ice bodies to meteorological forcing data have been set up and applied since many decades (e.g. Anderson, 1968; Kraus, 1975; Anderson, 1976; Kuhn, 1979; Male and Granger, 1981; Kuhn, 1987; Siemer, 1988; Morris, 1989, 1991; Munro, 1991). Studies have shown that the synthesis of these information provides a consistent understanding of the relevant mass and energy fluxes at the glacier-atmosphere interface, which in turn provides the necessary physical foundations to translate micro-meteorological conditions on glaciers into local melt rates (e.g. Sauter and Galos, 2016; Wagnon et al., 1999; Oerlemans, 2001; Mölg and Hardy, 2004; Obleitner and Lehning, 2004; Van Den Broeke et al., 2006; Reijmer and Hock, 2008; Mölg et al., 2008; Nicholson et al., 2013).

Distributed mass balance models combine the local melt information to a glacier-wide surface mass change information and thus offer the possibility to attribute and detect glacier mass and volume responses to changes in the climatic forcings (e.g. Klok and Oerlemans, 2002; Hock and Holmgren, 2005; Mölg et al., 2009; Sicart et al., 2011; Cogley et al., 2011). Although the accumulation and redistribution of snow are still deficient (e.g. Sauter et al., 2013), when coupled with atmospheric models such models have the potential to simulate present and future glacier evolution or to serve as a useful tool for monitoring climatic glacier mass change (Machguth et al., 2006). A well calibrated model is a suitable platform for sensitivity, detection and attribution analyses as well as a tool for the quantification of inherent uncertainties (e.g. Mölg et al., 2014; Maussion et al., 2015; Rye et al., 2012; Mölg et al., 2012; Sauter and Obleitner, 2015; Galos et al., 2017; van Pelt et al., 2012; Østby et al., 2017).

Over recent decades, several distributed mass balance models of varying complexity have been developed and successfully applied to different glacier systems and climate regimes. The models range from simple degree-day models (e.g. Radić and Hock, 2006; Schuler et al., 2005) to intermediate models (e.g. Machguth et al., 2009) and complex snow cover and glacier re-solving physical models (e.g. Bartelt and Lehning, 2002; Vionnet et al., 2012; Hock and Holmgren, 2005; Klok and Oerlemans, 2002; Sicart et al., 2011; Weidemann et al., 2018; Huintjes et al., 2015b; Mölg et al., 2009; Michlmayr et al., 2008; van Pelt et al., 2012). The latter model class is usually based on the same fundamental physical principles but differ in the parameter-isation schemes and implementation techniques. Different research groups have their own in-house solutions which are often extended and modified for specific scientific questions and studies. The fact that often several sub-versions of the same model exist, with some of them being not freely available, makes it difficult for users to having access to up-to-date software. Ideally, a platform should (i) be continuously maintained, (ii) provide newly developed parameterisations, (iii) compile different model subversions developed for specific research needs, (iv) be easily extensible and (v) be well documented and readable.

Here we present an open-source coupled snowpack and ice surface energy and mass balance model in Python (COSIPY) designed to meet these requirements. The structure is based on the predecessor model COSIMA (COupled Snowpack and Ice surface energy and MAss balance model, Huintjes et al., 2015b). COSIPY provides a lean, flexible and user-friendly framework

for modelling distributed snow and glacier mass changes. The framework consists of a computational core that forms the runtime environment and handles initialization, input-output (IO) routines, parallelization, and the grid and data structures. In most cases, the runtime environment does not require any changes by the user. To increase the user-friendliness, additional features are available, such as a restart option for operational applications and automatic comparison between simulation and ablation stakes. These features will be further refined in the future. Physical processes and parameterisations are handled separately by modules. The modules can be easily modified or extended to meet the needs of the end user. This structure provides maximum flexibility without worrying about internal numerical issues. The model is completely based on open-source libraries and is provided on a freely accessible git repository (https://github.com/cryotools/cosipy) for non-profit purpose. Scientists can actively participate in extending and improving the model code. Changes to the code are automatically tested with Travis CI (www.travis-ci.org) when uploaded to the repository. It is planned to publish updates in regular intervals. To make working with COSIPY easier, a community platform (https://cosipy.slack.com) has been set up in addition to a detailed readthedocs documentation (https://cosipy.readthedocs.io/en/latest), allowing users and developers to exchange experiences, report bugs and communicate needs.

In this work, we describe the physical basics, parameterisations and outline the numerical implementation of the model version COSIPY v1.3 (https://doi.org/10.5281/zenodo.3613921). Section 2 gives an overview of the model concept, followed by the description of the modules (Section 3). The model architecture and the input/output are explained in Section 4). Section 5 shows different applications of the model. The last section (Section 7) documents the code availability and software requirements.

## 2 Model concept

### 2.1 Fundamental equations

COSIPY is a multi-layered process resolving energy and mass balance model for the simulation of past, current and future glacier changes. The model is based on the concept of energy and mass conservation. The snow/ice layers are described by the volumetric fraction of ice $\theta_i$, liquid water content $\theta_w$ and air porosity $\theta_a$. Continuity constraints require that

$$\theta_i + \theta_w + \theta_a = 1. \tag{1}$$

The inherent properties, such as snow density $\rho_s$ or specific heat of snow $c_s$, follow from the volumetrically weighted properties of the constitutes. For example, snow density is related by

$$\rho_s = \theta_i \cdot \rho_i + \theta_w \cdot \rho_w + \theta_a \cdot \rho_a, \tag{2}$$

where $\rho_i$ is the ice density, $\rho_w$ the water density, and $\rho_a$ the air density (Bartelt and Lehning, 2002). Exchange processes at the surface, the energy release and consumption through phase changes, control the vertical temperature distribution within

the snow and ice layers. The energy balance also includes incoming shortwave radiation absorption and the sublimation or deposition of water vapour. Assuming the vertical temperature profile is given by $T_s(z,t)$, where $z$ is the depth, the energy conservation can be represented by

$$\rho_s(z,t)c_s(\theta,z,t)\frac{\partial T_s(z,t)}{\partial t} - k_s(\theta,z,t)\frac{\partial^2 T_s(z,t)}{\partial z^2} = Q_p(z,t) + Q_r(z,t) \tag{3}$$

where $c_s = c_i\,\theta_i + c_w\,\theta_w + c_p\,\theta_a$ and $k_s = k_i\,\theta_i + k_w\theta_w + k_a\theta_a$ are the volume-specific bulk heat capacity and bulk thermal conductivity of the snow cover (Bartelt and Lehning, 2002). Alongside the volume-specific heat capacity, COSIPY also offers the option of using empirical form $k_s = 0.021 + 2.5(\rho_s/1000.0)^2$ (Huintjes et al., 2015a). The first term on the right-hand side ($Q_p$) is the volumetric energy sink or source by melting and meltwater refreezing. The second term ($Q_r$) is the volumetric energy surplus by the absorption of shortwave radiation (see Eq. 13).

The exchange processes at the snow/ice-atmosphere interface control the surface temperature $T_s(z = 0,t)$ at an infinitesimal skin layer. From the energy conservation follows

$$k_s(\theta,z=0,t)\frac{\partial T_s(z=0,t)}{\partial z} = q_{sw} + q_{lw} + q_{sh} + q_{lh} + q_{rr}, \tag{4}$$

where $q_{sw}$ is the net-shortwave radiation energy, $q_{lw}$ is the net-longwave radiation energy, $q_{sh}$ is the sensible heat flux, $q_{lh}$ is the latent heat flux, and $q_{rr}$ is the heat flux from rain. To solve Eq. (4) for $T_s(z = 0,t)$, the fluxes on the right-hand side must
be parameterized (see Section 3). The parameterization results in a nonlinear equation which is solved iteratively. The left side of Eq. (4) provides the upper Neumann boundary condition (prescribed gradient) for Eq. (3). At the bottom of the domain, the temperature must be specified (Dirichlet boundary condition) by the user. The melting rates in the snow cover and glacier ice are derived diagnostically from the energy conservation by ensuring that the temperature does not exceed the melting point temperature $T_m$.

Eq. (4) is solved using a Limited Memory Broyden–Fletcher–Goldfarb–Shanno (BFGS) algorithm (Quasi-Netwon method) for bound constrained minimisation (Fletcher, 2000). Eq. (3) is then integrated with an implicit second-order central difference scheme (Ferziger and Perić, 2002). The heat sources can warm the snowpack and lead to internal melt processes. In case the liquid water content of a layer exceeds its irreducible water content (Coléou and Lesaffre, 1998; Wever et al., 2014),

$$\theta_e = \begin{cases} 0.0264 + 0.0099\,\dfrac{(1 - \theta_i)}{\theta_i}, & \text{if } \theta_i \le 0.23 \\ 0.08 - 0.1023\,(\theta_i - 0.03), & \text{if } 0.23 < \theta_i \le 0.812 \,, \\ 0, & \text{if } \theta_i > 0.812 \end{cases} \tag{5}$$

the excess water is drained into the subsequent layer (bucket approach). The liquid water is passed on until it reaches either a layer of ice or the glacier surface where it is considered to be runoff. For this purpose a threshold value was introduced which defines the transition from snow to ice. If no such layer exists, water is passed on until it reaches the lower limit of the domain

and is then considered as runoff. Meltwater refreezing and subsurface melting during percolation change the volumetric ice and water contents. Subsurface melt occurs when energy fluxes, e.g. penetrating shortwave radiation, warms the layer to physically inconsistent temperatures of $T_s > T_m$. Since physical constraints require that $T_s \leq T_m$, the energy surplus is used to melt the ice matrix. Melt takes place when $T_s > T_m$ and the liquid water content increases by

$$5 \quad \Delta\theta_w(z,t) = \frac{c_i(z,t)\theta_i(z,t)\rho_i(z,t)(T_s(z,t) - T_m)}{L_f \rho_w}, \tag{6}$$

where $L_f = 3.34 \times 10^5 JKg^{-1}$ is the latent heat of fusion (Bartelt and Lehning, 2002). Mass conservation requires that the mass gain of liquid water content equals the mass loss of the volumetric ice content, so that

$$\Delta\theta_i(z,t) = \frac{\rho_w \Delta\theta_w(z,t)}{\rho_i}. \tag{7}$$

The latent energy needed by the phase change is

$$10 \quad Q_p(z,t) = L_f \Delta\theta_i(z,t)\rho_i, \tag{8}$$

which is an heat sink because $\Delta\theta_i(z,t)$ is positive at melting. The energy used for melting ensures that $T_s(z,t)$ does not rise above $T_m$. In case $\theta_w > 0$ and $T_s < T_m$, refreezing can take place. Changes in volumetric fractions and the release of latent energy due to phase changes are treated equally. As the temperature difference must be negative due to the given constraints, it follows from Eq. (6), Eq. (7), and Eq. (8) that $Q_p$ becomes positive and latent heat release warms the layer.

Many of the quantities and fluxes in Eq. (3) and Eq. (4) are not measured directly and have to be derived via corresponding parameterizations. The next section describes the parameterizations implemented in COSIPY v1.3.

## 2.2 Discretization and computational mesh

To solve the underlying differential equations, the computing domain must be discretized. Since extreme gradients in temperature, density and liquid water content can develop in the snowpack, COSIPY uses a dynamic, non-equidistant mesh. The
20 mesh consists of so-called nodes that store the properties of the layers (e.g. temperature, density, and liquid water content), and is continuously adjusted during run-time by a re-meshing algorithm, i.e. the number and height of the individual layers vary with time. Currently, two algorithms are implemented: (i) A logarithmic approach, where the layer thicknesses gradually increase with depth by a constant stretching factor. Thus, layers close to the surface have a higher spatial resolution, which is advantageous for the computation of the energy and mass fluxes at the surface. Re-meshing is performed at each time step.
This is a fast method, but does not resolve sharp layering transitions, as these are smoothed by the algorithm. This approach is only recommended if a detailed resolution of the snow and ice cover is not required. (ii) An adaptive algorithm that assembles layers according to user-defined criteria. It uses density and temperature thresholds to determine when two successive layers are considered similar and when they are not. When both criteria are met, these layers are merged. Basically, the adaptive al-

gorithm runs in three consecutive steps: (1) adding/removing snow/ice at the surface, (2) adjusting the first layer, (3) updating internal layers.

(1) In the first step it is checked whether snow falls or melts away (note: internal layers can also melt). If snow falls on the glacier surface, it will only remain on the surface if it reaches a user-defined minimum snow thickness. Melt is removed from the first layer and all internal layers. After this step, layers can become very small and the thickness of the first layer no longer corresponds to the user-specified constant thickness. Therefore, it is necessary to re-mesh the layers.

(2) In the second step, the top layer is adjusted first. The top layer is re-meshed so that this layer always has the user-defined layer thickness (default value is 0.01 m). The adaptation of the top layers together with internal melting processes can reduce the internal layers to a very low thickness. To avoid thin layers, the layers are merged or split in the third step.

(3) In the third step, internal layers are splitted or merged. For each layer, a check is made to identify layers with a thickness of less than a defined minimum layer thickness. Such thin layers are merged with the layer below. Also if the differences in temperature and density of two subsequent layers are less than a user defined threshold (similarity criteria), they will be merged. How often a merging/splitting can take place per time step is also defined by the user (correction steps). Unlike CROCUS (Vionnet et al., 2012), internal re-meshing always starts from the surface, i.e. the uppermost layers are adapted first. Depending on how many correction steps are set by the user, it can happen that only the uppermost layers are re-meshed.

When two layers are merged, the liquid water content and the heights of the two layers are added and the new density is calculated from the volumetrically weighted densities of the two layers. To ensure energy conservation, the total energy is determined from the internal energies and converted into the new temperature. Unlike the logarithmic approach, adaptive re-meshing resolves individual layers but slightly increases both computing time and data volume.

# 3  Model physics and modules

## 3.1  Snowfall and precipitation

When snowfall is given, it is assumed that the data represents the effective accumulation since snowdrift and snow particle sublimation are not explicitly treated in the model. Otherwise, snowfall is derived from the precipitation data using a logistic transfer function. The proportion of solid precipitation smoothly scales between 100 % ($0\,°C$) and 0 % ($2\,°C$), as suggested by Hantel et al. (2000). The fresh snow density for the conversion into snow depth is a function of air temperature and wind velocity

$$\rho_s(z=0,t) = \max\left[ a_f + b_f(T_{z_t} - 273.16) + c_f u_{z_v}{}^{1/2}, \rho_{min} \right], \tag{9}$$

with the empirical parameters $a_f = 109\,\mathrm{kg\,m^{-3}}$, $b_f = 6\,\mathrm{kg\,m^{-3}\,K^{-1}}$, $c_f = 26\,\mathrm{kg\,m^{-7/2}\,s^{1/2}}$, and $\rho_{min} = 50\,\mathrm{kg\,m^{-3}}$ (Vionnet et al., 2012). In both cases fresh snow is only added when the height exceeds a certain user-defined threshold.

## 3.2 Albedo

The approach suggested by Oerlemans and Knap (1998) parametrizes the evolution of the broadband albedo. The decay of the snow albedo at a specific day depends on the snow age at the surface and is given by

$$\alpha_{snow} = \alpha_f + (\alpha_s - \alpha_f) \exp\left(\frac{s}{\tau^*}\right), \tag{10}$$

where $\alpha_s$ is the fresh snow albedo and $\alpha_f$ the firn albedo. The albedo time scale $\tau^*$ specifies how fast the snow albedo drops from fresh snow to firn. The number of days after the last snowfall is given by parameter $s$. Besides the temporal change, the overall snowpack thickness impacts the albedo. If the thickness of the snowpack $d$ is thin, the albedo must tend towards the albedo of ice $\alpha_i$. If one introduces a characteristic snow depth scale $d^*$ (e-folding) the full albedo can be written as

$$\alpha = \alpha_{snow} + (\alpha_i - \alpha_{snow}) \exp\left(\frac{-d}{d^*}\right). \tag{11}$$

The model resets the albedo to fresh snow, if the snow accumulation exceeds a certain threshold (default value is 0.01 m) within one time step. This approach neglects sudden short-term jumps in albedo, which can occur when thin fresh snow layers quickly melt away. To account for this effect, the age of the underlying snow is also tracked. If the fresh snow layer melts faster than $\tau^*$, the age of the snow cover is reset to the value of the underlying snow (Gurgiser et al., 2013).

## 3.3 Radiation fluxes

The net-shortwave radiation in the energy conservation equation Eq. (3) is defined as

$$q_{sw} = (1 - \alpha) \cdot q_G, \tag{12}$$

where $q_G$ is the incoming shortwave radiation, and $\alpha$ the snow/ice albedo. A proportion of the net shortwave radiation $q_{sw}$ can penetrate into the uppermost centimetres of the snow or ice (Bintanja and Van Den Broeke, 1995). The resulting absorbed radiation at depth $z$ is calculated with

$$Q_r(z,t) = \lambda_r \, q_{sw} \, \exp(-z\beta), \tag{13}$$

where $\lambda_r$ is the fraction of absorbed radiation (0.8 for ice; 0.9 for snow), and $\beta$ the extinction coefficient (2.5 for ice; 17.1 for snow). Physical constraints require that $T_s \leq T_m$ so that the energy surplus is used to melt the ice matrix (see Section 2).

In case the incoming longwave radiation $q_{lw_{in}}$ is observed, the net-longwave radiation is obtained by

$$q_{lw} = q_{lw_{in}} - \varepsilon_s \sigma T_0{}^4, \tag{14}$$

where $\varepsilon_s$ is the surface emissivity which is set to a constant close or equal to 1. In the absence of $q_{lw_{in}}$, the flux is parametrized by means of air temperature $T_{z_t}$ and atmospheric emissivity,

$$\varepsilon_a \; = \; \varepsilon_{cs}(1 - N^2) + \varepsilon_{cl} N^2, \tag{15}$$

using the Stefan-Boltzmann law. Here, $N$ is the cloud cover fraction, $\varepsilon_{cl}$ the emissivity of clouds which is set to 0.984 (Klok and Oerlemans, 2002), and $\varepsilon_{cs}$ the clear sky emissivity. The latter is given by

$$\varepsilon_{cs} \; = \; 0.23 + 0.433 \, (e_{z_t}/T_{z_t})^{1/8}, \tag{16}$$

where $e_{z_t}$ is the water vapor pressure (Klok and Oerlemans, 2002).

### 3.4 Turbulent fluxes

The turbulent fluxes, $q_{sh}$ and $q_{lh}$, in Eq. (4) are parametrized based on the flux-gradient similarity which assumes that the fluxes are proportional to the vertical gradient of state parameters. However, since meteorological parameters are only considered from one height in the model a bulk approach is used whereby the mean property between the measurement height and the surface is considered (e.g. Foken, 2008; Stull, 1988). Assuming that fluxes in the Prandtl layer are constant, dimensionless transport coefficients $C_H$ (Stanton number) and $C_E$ (Dalton number) can be introduced by vertically integrating the turbulent diffusion coefficients (Foken, 2008; Stull, 1988) so that the turbulent vertical flux densities can be written as

$$q_{sh} \; = \; \rho_a \, c_p \, C_H \, u_{z_v} \, (T_{z_t} - T_0) \tag{17}$$

$$q_{lh} \; = \; \rho_a \, L_v \, C_E \, u_{z_v} \, (q_{z_q} - q_0), \tag{18}$$

where $\rho_a$ is the air density (derived from the ideal gas law), $c_p$ is the specific heat of air for constant pressure, $L_v$ is the latent heat of vaporisation which is replaced by the latent heat of sublimation $L_s$ when $T_0 < T_m$, $u_{z_v}$ is the wind velocity at height $z_t$, $T_{z_t}$ and $q_{z_q}$ are the temperature and mixing ratio at height $z_t$ (assuming $z_t = z_q$), respectively, and $q_0$ is the mixing ratio at the surface where it is assumed that the infinite skin layer is saturated. Unlike the turbulent diffusion coefficients, the bulk coefficients are independent of the wind speed and only depend on the stability of the atmospheric stratification and the roughness of the surface. The aerodynamic roughness length $z_{0_v}$ is simply a function of time and increases linearly for snowpacks from fresh snow to firn (Mölg et al., 2012). For glaciers, $z_{0_v}$ is set to a constant value. According to the renewal theory for turbulent flow, $z_{0_q}$ and $z_{0_t}$ are assumed to be one and two orders of magnitude smaller than $z_{0_v}$, respectively (Smeets and van den Broeke, 2008; Conway and Cullen, 2013).

COSIPY provides two options to correct the flux-profile relationship for non-neutral stratified surface layers, by adding a stability correction using the (1) bulk Richardson-Number, and (2) Monin-Obukhov similarity theory (e.g. Conway and Cullen, 2013; Radić et al., 2017; Stull, 1988; Foken, 2008; Munro, 1989). Using the bulk Richardson number the dimensionless

transport coefficients can be written in the form

$$C_H = \frac{\kappa^2}{\ln\left(\dfrac{z}{z_{0_v}}\right)\ln\left(\dfrac{z}{z_{0_t}}\right)}\Psi_{Ri}(Ri_b) \tag{19}$$

$$C_E = \frac{\kappa^2}{\ln\left(\dfrac{z}{z_{0_v}}\right)\ln\left(\dfrac{z}{z_{0_q}}\right)}\Psi_{Ri}(Ri_b), \tag{20}$$

whereas the stability function

$$5 \quad \Psi_{Ri}(Ri_b) = \begin{cases} 1, & \text{if } Ri_b < 0.01 \\ (1 - 5\,Ri_b)^2, & \text{if } 0.01 \leq Ri_b \leq 0.2 \,, \\ 0, & \text{if } Ri_b > 0.2 \end{cases} \tag{21}$$

accounts for reduction of the vertical fluxes by thermal stratification and is a function of the Richardson number. The Richardson number

$$Ri_b = \frac{g}{T_{z_t}} \cdot \frac{(T_{z_t} - T_0)(z_t - z_{0_t})}{(u_{z_v})^2}, \tag{22}$$

follows from the turbulent kinetic energy equation and relates the generation of turbulence by shear and damping by buoy-
10 ancy (Stull, 1988). In the stable case ($0.01 \leq Ri_b \leq 0.2$), the function describes the transition from turbulent flow to a quasi-laminar non-turbulent flow, and hence, reduces the vertical fluxes. Once $Ri_b$ exceeds the critical value $Ri_b = 0.2$, turbulence eventually extinguishes, and the vertical exchange is suppressed.

According to the Monin-Obukhov similarity theory, atmospheric stratification can be characterised by the dimensionless parameter

$$15 \quad \zeta = z/L. \tag{23}$$

where

$$L = \frac{u_*^3}{\kappa \dfrac{g}{T_{z_t}} \dfrac{q_{sh}}{\rho_a \cdot c_p}} \tag{24}$$

is the so-called Obukhov length with $u_*$ is the friction velocity and $\kappa$ (0.41) the von Kármán constant (Stull, 1988; Foken, 2008). The length scale relates dynamic, thermal and buoyancy processes and is proportional to the height of the dynamic

sub-layer. The bulk aerodynamic coefficients for momentum $C_D$, heat $C_H$ and moisture $C_E$ for non-neutral conditions

$$C_D = \frac{\kappa^2}{\left[\ln\left(\frac{z}{z_{0_v}}\right) - \Psi_m(\zeta) - \Psi_m\left(\frac{z_{0_v}}{L}\right)\right]^2} \tag{25}$$

$$C_H = \frac{\kappa C_D^{1/2}}{\left[\ln\left(\frac{z}{z_{0_t}}\right) - \Psi_t(\zeta) - \Psi_t\left(\frac{z_{0_t}}{L}\right)\right]} \tag{26}$$

$$C_E = \frac{\kappa C_D^{1/2}}{\left[\ln\left(\frac{z}{z_{0_q}}\right) - \Psi_q(\zeta) - \Psi_q\left(\frac{z_{0_q}}{L}\right)\right]} \tag{27}$$

$$\tag{28}$$

can be derived by integrating the universal functions (Businger et al., 1971; Dyer, 1974) where

$$\Psi_m(\zeta) = \begin{cases} 2\ln\left(\frac{1+\chi}{2}\right) + \ln\left(\frac{1+\chi^2}{2}\right) - 2tan^{-1}\chi + \frac{\pi}{2} & \zeta < 0 \\ -b\zeta & 0 \le \zeta \le 1 \\ (1-b)(1+\ln\zeta) - \zeta & \zeta > 1 \end{cases}, \tag{29}$$

$$\Psi_t(\zeta) = \Psi_q(\zeta) = \begin{cases} \ln\left(\frac{1+\chi^2}{2}\right) & \zeta < 0 \\ -b\zeta & 0 \le \zeta \le 1 \\ (1-b)(1+\ln\zeta) - \zeta & \zeta > 1 \end{cases}, \tag{30}$$

with $\chi = (1-a\zeta)^{1/4}$, $a = 16$ and $b = 5$ are the stability-dependent correction functions. The computation of the stability functions requires an a priori assumption (Munro, 1989) about $L$ which in turn depends on $q_{sh}$ and the friction velocity

$$u_* = \frac{\kappa u_{z_v}}{\ln\left(\frac{z}{z_{0_v}}\right) - \Psi_m(\zeta)}. \tag{31}$$

COSIPY uses an iterative approach to resolve the dependency of these variables. At the beginning of the first iteration $u_*$ (Eq. 31) and $q_{sh}$ (Eq. 17) are approximated assuming a neutral stratification ($\zeta = 0$). These quantities are then used to calculate $L$ (Eq. 24). In the next iteration, the updated $L$ is then used to correct $u_*$ and $q_{sh}$ . The iteration is repeated until either the changes in $q_{sh}$ are less than $1 \cdot 10^{-2}$ or a maximum number of 10 iterations is reached. As already shown by other studies, the algorithm usually converges in less than 10 time steps (Munro, 1989).

## 3.5 Snow densification

Snow settling during metamorphism and compaction under the weight of the overlying snowpack generally increases the snow density over time (Anderson, 1976; Boone, 2004; Essery et al., 2013). The snow density is a key characteristic of the snowpack, which is used by COSIPY to derive important snow properties such as thermal conductivity and liquid water content. Assuming that a rapid settlement of fresh snow occurs simultaneously with slow compaction by the load resisted by the viscosity, the rate of density change of each snow layer becomes

$$\frac{1}{\rho_s(z,t)} \frac{d\rho_s(z,t)}{dt} = \frac{M_s(z,t)\,g}{\eta(z,t)} + c_1 \exp\left[-c_2(T_m - T_s) - c_3 \max\left(0, \rho_s(z,t) - \rho_0\right)\right], \tag{32}$$

with $M_s$ is the overlying snow mass, $c_1 = 2.8 \times 10^{-6}\,\mathrm{s}^{-1}$, $c_2 = 0.042\,\mathrm{K}^{-1}$, $c_3 = 0.046\,\mathrm{m}^3\,\mathrm{kg}^{-1}$, and the viscosity

$$\eta(z,t) = \eta_0 \exp\left[c_4(T_m - T_s) + c_5\rho_s\right] \tag{33}$$

where $\eta_0 = 3.7 \times 10^7\,\mathrm{kg\,m^{-1}\,s^{-1}}$, $c_4 = 0.081\,\mathrm{K}^{-1}$, and $c_5 = 0.018\,\mathrm{m}^3\,\mathrm{kg}^{-1}$ (Anderson, 1976; Boone, 2004; Essery et al., 2013).

## 3.6 Mass changes

The total mass changes may be written as the integral expression

$$\frac{\partial}{\partial t}\int_0^d \rho_s\,dz = \frac{\partial}{\partial t}\int_0^d \theta_i(z,t)\rho_i\,dz + \frac{\partial}{\partial t}\int_0^d \theta_w(z,t)\rho_w\,dz + \frac{\partial}{\partial t}\int_0^d \theta_a(z,t)\rho_a\,dz, \tag{34}$$

which follows directly from Eq. (2). So any net mass change must be accompanied by changes in ice fraction, liquid water content, and porosity within the snow/ice column of height $d$. The continuity equation for ice fraction (first term on the right side) may be written as

$$\frac{\partial}{\partial t}\int_0^d \theta_i(z,t)\rho_i\,dz = \frac{\partial}{\partial t}\int_0^d \Delta\theta_i(z,t)\rho_i\,dz + SF - \frac{q_m}{L_f} + \frac{q_{lh}}{L_s} + \frac{q_{lh}}{L_v}, \tag{35}$$

where the integral on the right side describes the internal mass changes by melt and refreezing, $SF$ the mass gain by snowfall, and $q_m/L_f$ is the mass loss by melt. The last two terms of this equation, $q_{lh}/L_s$ and $q_{lh}/L_v$, are the sublimation/deposition and evaporation/condensation fluxes at the surface, respectively, depending on the sign of $q_{lh}$ and $T_s(z=0,t)$. Melt energy $q_m$ is the energy surplus at the surface which is available for melt, and follows from Eq. (4). Similarly, we can extend the continuity

equation for the liquid water content which reads as

$$\frac{\partial}{\partial t}\int\limits_{0}^{d}\theta_w\rho_w(z,t)\,dz = \frac{\partial}{\partial t}\int\limits_{0}^{d}\Delta\theta_w(z,t)\rho_w\,dz + R_f + \frac{q_m}{L_f} + \frac{q_{lh}}{L_v} - Q \tag{36}$$

with the integral on the right side describing the internal mass changes of liquid water by melt and refreezing, $R_f$ the mass gain by liquid precipitation, and $Q$ the runoff at the bottom of the snowpack. COSIPY calculates all terms and writes them to the output file.

## 4    Model architecture

Basically, COSIPY consists of a model kernel which is extended by modules. The model kernel forms the underlying model structure and provides the IO routines, takes over the discretisation of the computational mesh, parallelizes the simulations, and solves the fundamental mass- and energy conservation equations Eq. (2) and Eq. (3). These tasks are independent of the implementations of the parametrization and usually, do not require any modification by the end-user.

COSIPY is a one-dimensional model that resolves vertical processes at a specific point on the glacier. For spatially distributed simulations, the point model is integrated independently at each point of the glacier domain, neglecting the lateral mass and energy fluxes. The independency of the point models simplifies scaling for larger computer architectures, which led to the COSIPY model architecture being designed for both local workstations and High-Performance Computing Cluster (HPCC). So far, the model has been successfully tested on Slurm Workload Manager (https://slurm.schedmd.com) and PBS job scheduling systems (https://www.pbspro.org). Regardless of whether the distributed simulations are integrated on a single-core or multi-core computing environment, the point model sequence is always the same. During initialisation, the atmospheric input data is read in, and the mesh is generated. With distributed spatial simulations, the data is distributed across the available cores, and one-dimensional calculations are performed for each grid point.

At the beginning of each time step, it is checked whether snowfall occurs and must be added to the existing snow cover. Subsequently, the computational mesh is re-meshed to ensure numerical stability. Afterwards, the albedo (Eq. 11) and the roughness length are updated. Once these steps have been performed, the heating and melting of snow by penetrating short-wave radiation (Eq. 13, 6, and 7) is determined and the surface energy fluxes and surface temperature (Eq. 4) are derived. The resulting meltwater, both from surface and subsurface melt, is then percolated through the layers (bucket approach). Next the heat equation (Eq. 3) is solved after all terms on the right side have been determined.

### 4.1    Input and Output (IO) and initial condition

The model is driven by meteorological data that must be provided in a corresponding NetCDF file (see https://cosipy.readthedocs. io/en/latest/Ressources.html). Input parameters include atmospheric pressure, air temperature, cloud cover fraction, relative humidity, incoming shortwave radiation, total precipitation and wind velocity. Optional snowfall and incoming longwave ra-

diation can be used as forcing parameters. If the snow height (or snow water equivalent) and/or surface temperature are also specified in the input file, these are used as initial conditions. Otherwise, snow depth and surface temperature are assumed to be homogeneous in space at the start of the simulation according to the specifications in the configuration file. In addition to meteorological parameters, COSIPY requires static information such as topographic parameters and a glacier mask. Example
workflows for creating and converting static and meteorological data into the required NetCDF input is included in the source code (https://cosipy.readthedocs.io/en/latest/Documentation.html#quick-tutorial). Besides the standard output variables, there is also the possibility to store vertical snow profile information, although to save memory we can only recommend this for single point simulations. To reduce the amount of data, the users can specify which of the output variables will be stored.

## 5 Model applications

### 5.1 Zhadang glacier, High Mountain Asia

The first example shows the application of COSIPY to the Zhandang glacier, which is located on the north-eastern slope of the Nyainqentanglha Mountains (30°28.2'N, 90°37.8'E) on the Central Tibetan Plateau.

### 5.1.1 Single-site simulation

For single-site simulation, we use hourly data from May 2009 to June 2012 from an automatic weather station (AWS) on the
15 Zhadhang Glacier (Huintjes et al., 2015b). The relevant variables air pressure $p_{z_t}$, air temperature $T_{z_t}$, relative humidity $RH_{z_t}$, incident short-wave radiation $q_G$, snowfall $SF$ and wind speed $u_{z_v}$ were measured by the AWS. Due to the harsh and remote environment, the time series show gaps that were filled with the High Asia Refined Analysis (HAR; Maussion et al., 2014) product. The cloud cover fraction $N$ was provided by ERA5 (Hersbach and Dee, 2016) data. We compare the simulated snow temperature $T_s$ and surface height change $\Delta H$ with the AWS measurements. Furthermore, ablation stakes were installed on
the glacier to determine the loss of mass at various locations on the glacier. A detailed description of the data, the AWS sensors used, the post-processing procedure and the discussion can be found in Huintjes et al. (2015b) and Huintjes (2014).

**Table 1.** Observed and simulated ice ablation (mm w.e.) for three periods at the automatic weather station on the Zhadang glacier

| Period | 13.07.2009-30.08.2009 | | 17.05.2010-10.09.2010 | | 26.07.2011-16.08.2011 | |
| --- | --- | --- | --- | --- | --- | --- |
| | total | per day | total | per day | total | per day |
| Stake | 1072 | 22 | 2255 | 19 | 150 | 7 |
| Simulated | 1190 | 25 | 2150 | 19 | 160 | 8 |

**Simulation**. Figure 1a and 1b show the glacier surface temperatures determined from longwave radiation measurements and from COSIPY simulations for two periods where in-situ measurements are available. The model represents both the daily variability ($R^2$ = 0.83, p-values < 0.001) and the magnitude of the observed surface temperature. The root mean square error
is 3.3 K and 2.2 K for the two periods, respectively, and is thus within the typical uncertainty range of long-wave radiation

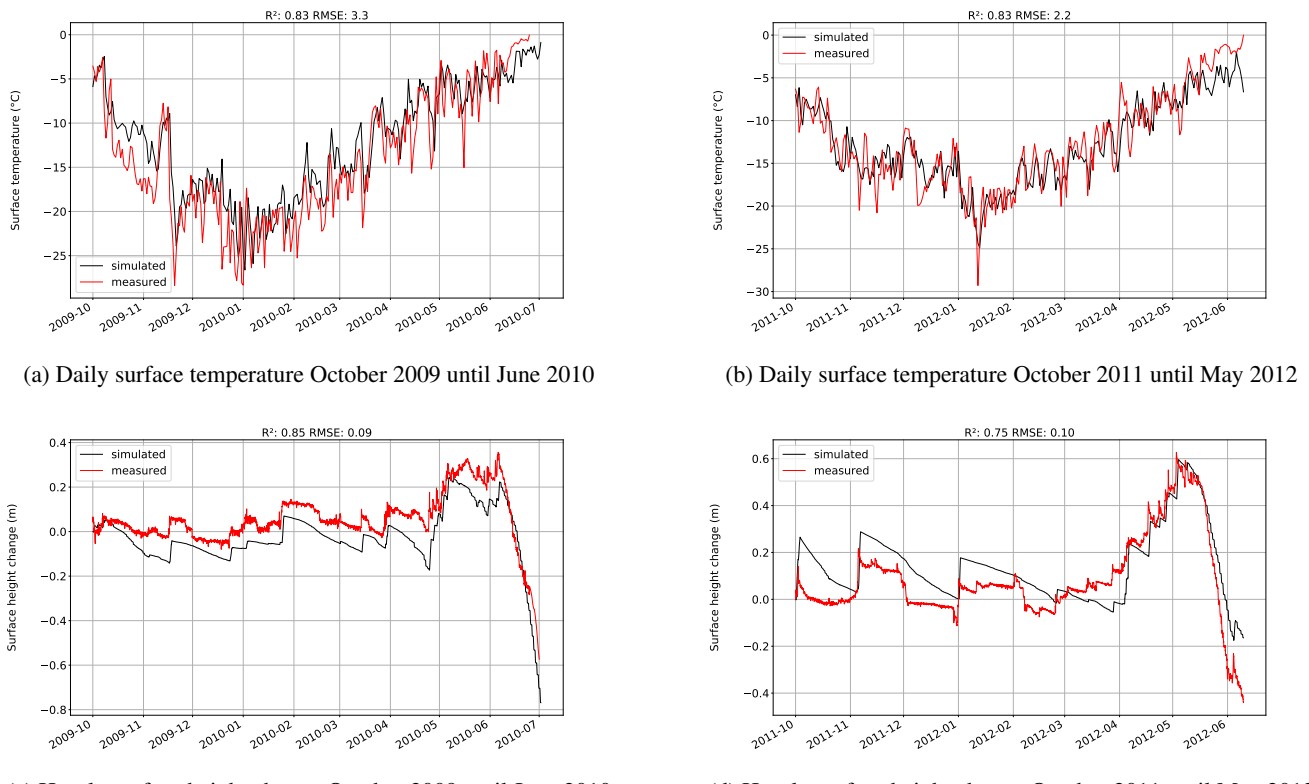

(a) Daily surface temperature October 2009 until June 2010

(b) Daily surface temperature October 2011 until May 2012

(c) Hourly surface height change October 2009 until June 2010

(d) Hourly surface height change October 2011 until May 2012

**Figure 1.** Simulated and measured surface temperatures and surface height changes (in both cases permanent snow cover) at the location of the automatic weather station at the Zhadang glacier.

measurements. The simulated cumulative mass balance over the entire period from April 2009 to May 2012 is -2.9 m w.e. Figure 1c and 1d show the simulated and measured $\Delta H$ for the two periods October 2009 to June 2010 and October 2011 to May 2012 where measurements are available. The daily and seasonal variability is well captured by the model ($R^2 = 0.85$, p-value < 0.001 and $R^2 = 0.75$, p-value < 0.001), even if snowfall seems to be too low during the first period. Nevertheless, overall the differences are consistently small with a RMSE of 0.09 m and 0.10 m. Table 1 shows the observed and simulated ice ablation for three different periods for which measurements are available. For all three periods, a high degree of agreement is evident, which reveals that the energy fluxes are represented by COSIPY.

### 5.1.2 Distributed simulation, scalability

For a distributed glacier-wide run we drive COSIPY by ERA5 data instead of in-situ observations. The ERA5 temperature and humidity data were interpolated across the topography using empirical lapse rates. Atmospheric pressure has been corrected using the barometric formula. The radiation model of Wohlfahrt et al. (2016) was used for the incoming shortwave radiation to account for effects of shadowing, slope and aspect. Total precipitation, cloud cover and horizontal wind velocity were used

directly from the closest ERA5 grid point (cf. Table 2). The computational domain consisted of 1837 grid cells with a spatial resolution of approximately 30 m (1 arcsecond) (see Fig. 2).

**Table 2.** COSIPY forcing variables and applied downscaling approaches for distributed simulation.

| Variable | Downscaling ERA5 data to elevation of the glacier | Applied approach for distributed fields on the glacier |
|---|---|---|
| Air pressure $p_{z_t}$ | Barometric formula | Barometric formula |
| Air temperature $T_{z_t}$ | Lapse rate | Lapse rate |
| Cloud cover fraction $N$ | - | - |
| Incoming shortwave radiation $q_G$ | - | Radiation modelling (Wohlfahrt et al., 2016) |
| Relative humidity $RH_{z_t}$ | Lapse rate | Lapse rate |
| Total precipitation $RRR$ | - | - |
| Wind speed $u_{z_v}$ | - | - |

**Simulation**. The glacier-wide cumulative surface mass balance for the decade 2009 to 2018 is presented in Figure 2. The simulated annual mass balance of for this period was -1.9 $m\ w.e.a^{-1}$. The results are in line with the analysis of Qu et al. (2014) who reported negative mass balances of $-1.9$, $-2.0$, $-0.8$ and $-2.7\ m\ w.e$ for the years 2009 to 2012. Furthermore, COSIPY reproduced the spatial distribution at different locations in the ablation area of the Zhadang glacier (cf. S1, S2 and S3 in Figure 2b)

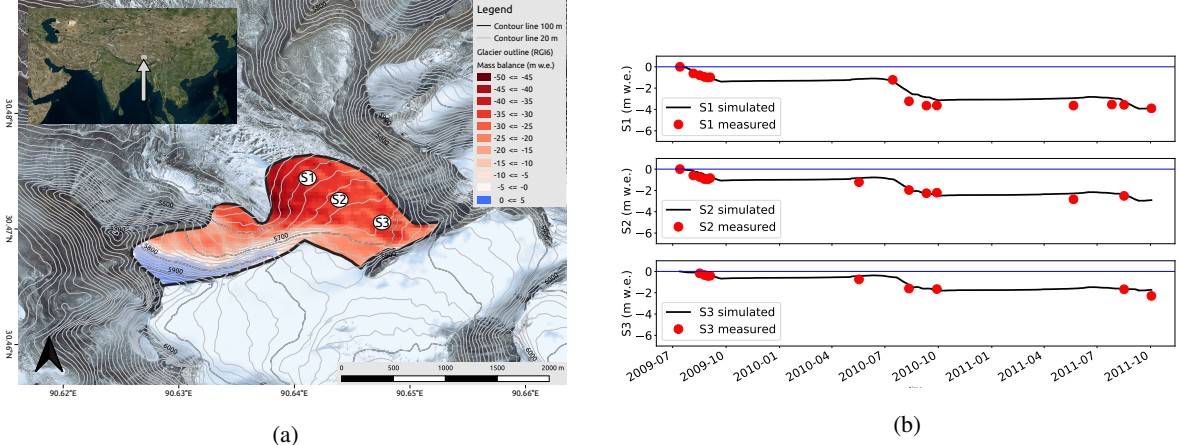

(a)                                                                 (b)

**Figure 2.** Distributed mass balance simulation of the Zhadang glacier. **(2a)** Cumulative climatic mass balance from 2009 to 2018 with 1827 grid points, contour lines (SRTM), glacier outline from Randolph Glacier Inventory 6.0 and a topographic map from Bing Maps (Microsoft, 2020); **(2b)** comparison of three measurements (ablation stakes) from July 2009 to October 2011 with the simulated cumulative surface mass balance of the corresponding grid point.

**Scalability**. A big challenge for large applications is usually the computational cost. To achieve the required performance, models should be scalable on parallel high-performance computing environments. For the model performance analysis, we use

a cluster with identical nodes, each consisting of two Intel Xeon(R) E5-2640 v4 CPUs operating at 2.4 GHz and connected via InfiniBand. Each processor has ten cores, 32 GB memory and a memory bandwidth of 68.3 GB/s. To test the performance of the parallelized COSIPY version, we performed a spatial simulation of the Zhadhang glacier. We used a 3 arcsecond ($\sim$ 90 m) Shuttle Radar Topography Mission (SRTM) terrain model so that the computational grid consists of 206 points. The

performance of the parallel version was then compared to the single-core solution by measuring the required execution time for different core setups (1-220 cores). Figure 3 shows the speedup compared to the single-core version, i.e. the ratio of the original execution time (single core) with the execution time of the corresponding node test (multiple cores). If the model is executed with 20 cores, the speedup is $\sim 2$. With 120 cores a speedup of $\sim 10$ is reached, i.e. each core has to calculate a maximum of two grid points. A speedup of more than $\sim 16$ is not possible with this system and is achieved with a number of

220 cores (more cores than grid points). The computation time is less than 35 minutes for a ten year period (hourly resolution) when using 220 cores. At this point, it should be mentioned that the performance can vary significantly on other HPCC systems and simulation conditions and should always be checked before submitting larger simulations to the cluster.

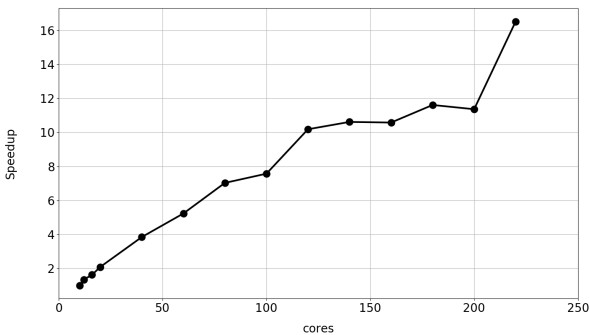

**Figure 3.** Speedup (execution time of single-core simulation divided by execution time of the corresponding multiple-core simulation) for computing a 10-year distributed COSIPY run on Zhadang glacier with 206 grid points.

## 5.2    Distributed mass- and energy balance simulation and operational application at Hintereisferner in the Austrian Alps

The Hintereisferner (HEF) is a valley glacier located in the Ötztal Alps of Austria (46.79°N, 10.74°E). The glacier begins high on the flank of the mountain Weißkugel, at approximately 3720 m, and runs down to its terminus at approximately 2460 m. HEF is a prime location for meteorological and glaciological research activities due to its monitoring infrastructure. There is a network of 4 automatic weather stations (AWS) and 4 precipitation gauges operated on, and in the vicinity of HEF. Since 2016, the University of Innsbruck is also running a permanent Terrestrial Laser Scanner (TLS) and a 5 m meteorological flux tower.

Measurement data is hourly transmitted to a data server. COSIPY is now being used to develop an operational mass balance prediction system for the 'Hintereisferner'. The model is driven by the latest COSMO2 analysis and forecast data (see Fig. 4).

With the forecast data the energy and mass flows on the glacier are predicted for the next 24 hours with a horizontal resolution of 30 m. The simulated fields are automatically visualised and provided on a web server. In the future the TLS measurements will be used to improve the forecast continuously. The system is currently running in test mode but will be available to the public in spring 2021.

In addition to the energy and mass flows at the surface, the snow/ice profiles will be stored. This will allow to compare the results with snow pits and to test the implementation of different parameterizations.

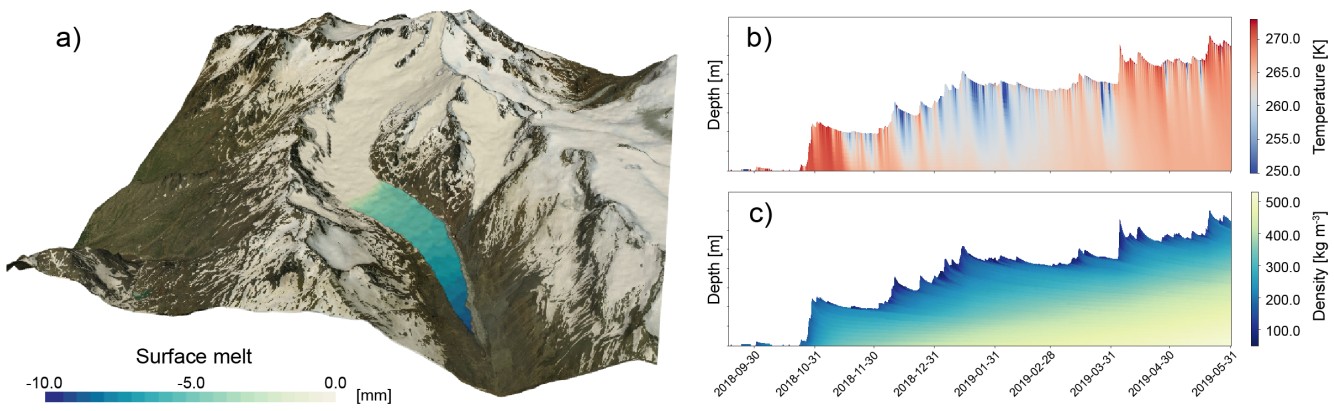

**Figure 4.** Operational application of COSIPY for the Hintereisferner. Panel a) shows the forecast of surface melt for 22 June 2020 based on COSMO2 data. Panels b) and c) show an example of the temperature and density profile for one site on the glacier tongue for the period September 2018 to June 2019.

### 5.3    Model intercomparison - Earth System Model-Snow Model Intercomparison Project

Within the Earth System Model-Snow Model Intercomparison Project (ESM-SnowMIP, Krinner et al., 2018) several of snow models were compared to evaluate different snow schemes and to improve the coupling of land surface snow models in Earth
System models. Ménard et al. (2019) describes the standardized input and evaluation data. Ten different sites representing mountainous regions (Europe and western USA), boreal forests (Canada), the Arctic (Finland) and urban regions (Japan) for periods between seven and 20 years (hourly resolution) are provided, including meteorological classification and details on measuring instruments and data processing. These quality controlled data are freely available on a PANGAEA repository (Ménard and Essery, 2019) and provide the possibility to benchmark new model developments, to detect uncertainties and to
reduce model errors. Unlike most of the models participating in the Intercomparison project, COSIPY is not a pure snow model, but still all necessary forcing variables are available to apply the model to the different test data sets. We downscaled wind speed from 10 to 2 m above ground using the logarithmic wind profile and calculated the relative humidity from the specific humidity using the saturation mixing ratio and water vapour. The simulated abledo, snow water equivalent (SWE) and snow depth were compared with the evaluation data offered on the online repository (https://doi.pangaea.de/10.1594/PANGAEA.897575).

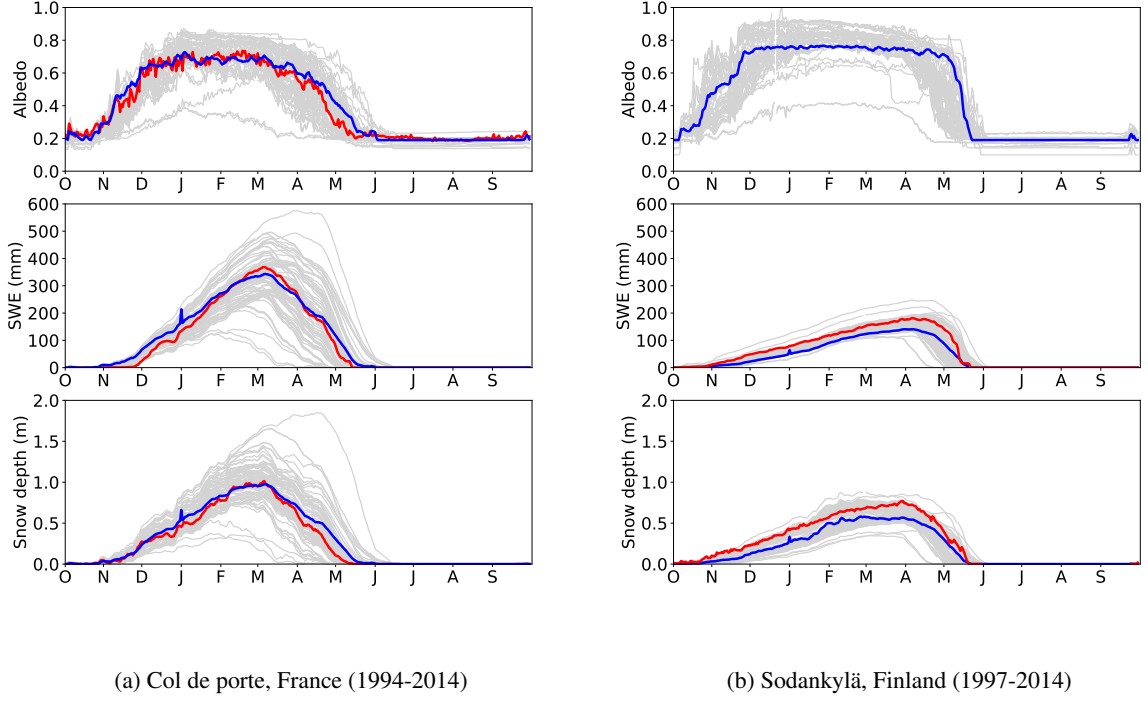

(a) Col de porte, France (1994-2014)  (b) Sodankylä, Finland (1997-2014)

**Figure 5.** Comparison of long-term daily mean COSIPY with ESM-SnowMIP simulations for two sites. COSIPY simulations (blue lines), measurements (red lines) and ESM-SnowMIP (grey lines) simulations of albedo, snow water equivalent (SWE) and snow depth at two sites. Measurements and simulations provided by Krinner et al. (2018).

Surface and soil temperature could not be compared, because no soil scheme is implemented in COSIPY which allows for warm surface and underground temperatures above the melting point. Figure 5 shows the daily long-term mean values of albedo, SWE and snow depth for two example sites. The ableldo parametrization was calibrated to fit the observed values at Col the porte. With the calibrated albedo parameterisation, COSIPY can reproduce the observed long-term snowpack evolution.
5  The results for Sodankylä, Finnland (cf. 5b) show a little lower snowpack compared to the measurements. The COSIPY runs for both sites are in the range of the ESM-SnowMIP ensemble simulations (see Figure 5, Krinner et al., 2018).

## 6  Conclusions

COSIPY provides a lean, flexible and user-friendly framework for modelling distributed snow and glacier mass changes. It provides a suitable platform for sensitivity, detection and attribution analyses as well as a tool for the quantification of
10  inherent uncertainties in mass balance studies. The model has a modular structure and allows the exchange of routines or parameterizations of individual physical processes with little effort. This structure allows the end user to quickly adapt the

model to their needs. The open design of COSIPY is well documented, and the modular approach allows a joint community-driven further development of the model in the future. In order to increase user-friendliness, additional functions are available, such as a restart option for operative applications and an automatic comparison between simulation and ablation data. These functions will be further refined in the future.

The model is written in Python and completely based on open source libraries. The model, source code, case studies and codes examples for data preprocessing are provided on a freely accessible Git repository (https://github.com/cryotools/cosipy) for non-profit purposes. The aim is to set up a community platform where scientists can actively participate in extending and improving the model code. To ensure quality control of the model code, changes to the code are automatically tested with Travis CI (www.travis-ci.org) when they are uploaded to the repository. It is planned to release updates at regular intervals. To make working with COSIPY easier, a community platform (https://cosipy.slack.com) has been set up in addition to readthedocs documentation (https://cosipy.readthedocs.io/en/latest), which allows users and developers to share experiences, report bugs and communicate needs.

Future improvements of COSIPY are expected by applying the model in different climates and varying topographical settings. Additional processes affecting the climatic mass balance of glaciers such as debris cover and snowdrift can be considered in further developments of the model. On the long run, one of the priorities will be to create a multiphysics environment that allows ensemble runs. In principle it is already possible to create ensemble simulations with different physical parameterizations and solvers, but COSIPY is not yet an ensemble multiphysics modelling environment. As a vision for the future it is conceivable to extend COSIPY for automatic ensemble simulations. So far, it is only possible to run COSIPY with different combinations of physical parameterizations or input uncertainties and then evaluate the statistics.

## 7   Code availability, documentation, and software requirements

COSIPY is based on the Python 3 language and is provided on a freely accessible git repository (https://github.com/cryotools/cosipy, last access: June 20, 2020). COSIPY can be used for non-profit purposes under the GPLv3 license (http://www.gnu.org/licenses/gpl-3.0.html). Scientists can actively participate in model development. A documentation with a sample workflow, information about input/output formats and the code structure is available under 'Read the Docs' (https://cosipy.readthedocs.io/en/latest/index.html, last access: June 20, 2020). As a community platform and user support, we use the groupware Slack (https://cosipy.slack.com, last accessed: June 20, 2020). The various official model releases will be registered with a unique DOI on Zenodo (https://doi.org/10.5281/zenodo.2579668, last access: June 20, 2020). For the result of this publication the version v1.3 (https://doi.org/10.5281/zenodo.3902191) was used. Each commit will be automatically tested with different Python 3 releases on Travis (https://travis-ci.org/cryotools/cosipy, last accessed June 20, 2020). The tested code coverage is tracked on CodeCov (https://codecov.io/github/cryotools/cosipy/, last accessed June 20, 2020). Since we have just started writing the tests, code coverage of 35 % is still low but will be increased in the near future. With the exception of the pre-processor for creating the static file (currently not working on Windows systems) the model should work on any operating system with Python 3 installed. However, support for operating systems other than Linux-based systems is limited because we develop and run

COSIPY exclusively on Linux-based systems. COSIPY is built on the following open-source libraries: numpy (van der Walt et al., 2011), scipy (Virtanen et al., 2019), xarray (Hoyer and Hamman, 2017), distribued, dask_jobqueue (Dask Development Team, 2016), and netcdf4 (https://doi.org/10.5281/zenodo.2669496).

# Appendix A: List of symbols

| Constant | Description | Unit | Default value |
|---|---|---|---|
| $c_p$ | specific heat of air | $J\ kg^{-1}\ K^{-1}$ | 1004.67 |
| $c_i$ | specific heat of ice | $J\ kg^{-1}\ K^{-1}$ | 2050.0 |
| $c_w$ | specific heat of water | $J\ kg^{-1}\ K^{-1}$ | 4217.0 |
| $d^*$ | albedo depth scale | $cm$ | 3.0 |
| $g$ | gravitational acceleration | $m\ s^{-2}$ | 9.81 |
| $k_a$ | thermal conductivity of air | $W\ m^{-1}\ K^{-1}$ | 0.026 |
| $k_i$ | thermal conductivity of ice | $W\ m^{-1}\ K^{-1}$ | 2.25 |
| $k_w$ | thermal conductivity of water | $W\ m^{-1}\ K^{-1}$ | 0.6089 |
| $Pr$ | turbulent Prandtl number | – | 0.8 |
| $T_m$ | melting point temperature | $K$ | 273.16 |
| $\alpha_s$ | fresh snow albedo | – | 0.9 |
| $\alpha_f$ | firn albedo | – | 0.55 |
| $\alpha_i$ | ice albedo | – | 0.3 |
| $\varepsilon_s$ | surface emissivity | – | 0.99 |
| $\eta_0$ | snow viscosity | $kg\ m^{-1}\ s^{-1}$ | $3.7 \times 10^7$ |
| $\kappa$ | von Kármán constant | – | 0.41 |
| $\rho_a$ | air density | $kg\ m^{-3}$ | 1.1 |
| $\rho_w$ | water density | $kg\ m^{-3}$ | 1000.0 |
| $\rho_i$ | ice density | $kg\ m^{-3}$ | 917.0 |
| $\rho_0$ | snow compaction parameter | $kg\ m^{-3}$ | 150.0 |
| $\sigma$ | Stefan–Boltzmann constant | $W\ m^{-2}\ K^{-4}$ | $5.67 \times 10^8$ |
| $\tau^*$ | albedo time scale | $days$ | 22 |

| Variable | Description | Unit |
|---|---|---|
| $c_s$ | specific heat of snow | $J\ kg^{-1}\ K^{-1}$ |
| $C_D$ | bulk transfer coefficient for momentum | – |
| $C_E$ | bulk transfer coefficient for latent heat | – |
| $C_H$ | bulk transfer coefficient for sensible heat | – |
| $e_{z_t}$ | water vapour pressure at height $z_t$ | $Pa$ |
| $Ew_{z_t}$ | saturation water vapour at height $z_t$ | $Pa$ |
| $Ew_{z_{0_t}}$ | saturation water vapour at the surface | $Pa$ |

| | | |
|---|---|---|
| $k_s$ | thermal conductivity of snow | $W\ m^{-1}\ K^{-1}$ |
| $L$ | Obukhov length | $m$ |
| $L_s$ | latent heat of sublimation | $J\ kg^{-1}$ |
| $L_f$ | latent heat of fusion | $J\ kg^{-1}$ |
| $L_v$ | latent heat of vaporisation | $J\ kg^{-1}$ |
| $ME$ | available melt energy | $W\ m^{-2}$ |
| $M_s$ | overlying mass | $kg$ |
| $N$ | cloud cover fraction | $-$ |
| $q_{lw}$ | net longwave radiation | $W\ m^{-2}$ |
| $q_{lw_{in}}$ | incoming longwave radiation | $W\ m^{-2}$ |
| $q_{lw_{out}}$ | outgoing longwave radiation | $W\ m^{-2}$ |
| $q_{sw}$ | net shortwave radiation | $W\ m^{-2}$ |
| $q_{sh}$ | sensible heat flux | $W\ m^{-2}$ |
| $q_{lh}$ | latent heat flux | $W\ m^{-2}$ |
| $q_{rr}$ | heat flux from rain | $W\ m^{-2}$ |
| $q_m$ | melt energy | $W\ m^{-2}$ |
| $q_0$ | mixing ratio at the surface | $kg\ kg^{-1}$ |
| $q_{z_q}$ | mixing ratio at height $z_q$ | $kg\ kg^{-1}$ |
| $p_{z_t}$ | air pressure at height $z_t$ | $hPa$ |
| $Q$ | runoff | $mw.e.$ |
| $Q_p$ | volumetric energy sink/source by melting and refreezing | $W\ m^{-3}$ |
| $Q_r$ | volumetric energy surplus by absorption of shortwave radiation | $W\ m^{-3}$ |
| $RH_{z_t}$ | relative humidity at height $z_v$ | $\%$ |
| $Ri_b$ | Bulk Richardson number | $-$ |
| $SF$ | snowfall | $m$ |
| $T_s$ | snow temperature | $K$ |
| $T_v$ | virtual air temperature | $K$ |
| $T_{z_t}$ | air temperature at height $z_t$ | $K$ |
| $T_0$ | surface temperature | $K$ |
| $u_{z_v}$ | wind speed at height $z_v$ | $m\ s^{-1}$ |
| $u_*$ | Friction velocity | $m\ s^{-1}$ |
| $z_t$ | measurement height of temperature | $m$ |
| $z_q$ | measurement height of humidity | $m$ |
| $z_t$ | measurement height of wind velocity | $m$ |
| $z_{0_v}$ | aerodynamic roughness length | $m$ |

| $z_{0_t}$ | roughness length for temperature | $m$ |
|---|---|---|
| $z_{0_q}$ | roughness length for specific humidity | $m$ |
| $\alpha$ | snow/ice albedo | — |
| $\varepsilon_{cl}$ | emissivity of clouds | — |
| $\varepsilon_{cs}$ | clear sky emissivity | — |
| $\varepsilon_a$ | total atmospheric emissivity | — |
| $\eta$ | snow viscosity | $kg\ m^{-1}\ s^{-1}$ |
| $\theta_w$ | liquid water content | — |
| $\theta_a$ | air porosity | — |
| $\theta_i$ | volumetric ice fraction | — |
| $\theta_e$ | irreducible water content | — |
| $\Theta$ | local slope | $^\circ$ |
| $\lambda_r$ | fraction of absorbed radiation | — |
| $\rho_s$ | snow density | $kg\ m^{-3}$ |
| $\Psi_{Ri}$ | stability function based on the Richardson-Number | — |
| $\Psi_m$ | stability function for momentum based on the Monin-Obukhov similarity theory | — |
| $\Psi_t$ | stability function for heat based on the Obukhov length | — |
| $\Psi_q$ | stability function for moisture based on the Obukhov length | — |

*Author contributions.* TS and CS are the initiators of the model. TS developed the model design and wrote most of the sections on the physical and numerical principles of the model. AA developed parts of the parameterizations and core classes, applied the model to the Zhadang glacier and the two sites of the ESM-SnowMIP. He also wrote the sections about the corresponding applications and code availability. TS did the simulations for the Hintereisferner. AA and TS were equally involved in the development of the documentation and maintenance of the Community Platform (Slack). During the entire development process, all authors discussed the individual steps of model development, the results and the structure of the manuscript.

*Competing interests.* The authors declare that they have no conflict of interest.

*Acknowledgements.* We gratefully acknowledge financial support by the Deutsche Forschungsgemeinschaft (DFG) with its project 'The impact of the dynamic and thermodynamic flow conditions on the spatio-temporal distribution of precipitation in southern Patagonia' (grant no. SA 2339/4-1) and 'Snow Cover Dynamics and Mass Balance on Mountain Glaciers' (grant no. SA 2339/7-1). Part of this work and the position of co-author Anselm Arndt was financed through the German Research Foundation's (DFG) research grant 'Precipitation patterns, snow and glacier response in High Asia and their variability on sub-decadal time scales, sub-project: snow cover and glacier energy and

mass balance variability' (prime-SG, SCHN 680/13-1). The development of the earlier version of the software and the field data used in this study were financed by the projects 'Dynamic response of glaciers of the Tibetan Plateau' (Dyn RG TiP, grant nos. SCHN 680/3-1, SCHN 680/3-2, SCHN 680/3-3) of DFG's Priority Programme 1372 'Tibetan Plateau: Formation—Climate—Ecosystems' (TiP) and the German Federal Ministry of Education and Research's (BMBF) programme 'Central Asia Monsoon Dynamics and Geo-Ecosystems'

5   (CAME), project 'Variability and Trends of Water-Budget Components in Benchmark Catchments of the Tibetan Plateau' (WET, grant no. 03G0804E). We would like to thank Eva Huintjes for her dedication to the fieldwork and for her contributions to the earlier versions of the software. Further, we acknowledge the efforts of all researchers and technicians from the involved German institutions and the Institute of Tibetan Plateau Research of the Chinese Academy of Sciences (CAS) for fieldwork at Zhadang Glacier. We specifically acknowledge the contribution of Yang Wei, Yao Tandong and Kang Shichang in this respect. We gratefully thank Richard Essery and Gerhard Krinner for

10   providing the data of the ensemble simulations for two sites of ESM-SnowMIP. We also wish to thank David Loibl for his contribution in the form of logo, ideas and discussions regarding programming strategies, and for hosting COSIPY on his platform https://cryo-tools.org/. We would also like to thank Samuel Morin and the anonymous reviewer for their constructive comments and ideas.

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
