# Peer review of "COSIPY v1.3 - An open-source coupled snowpack and ice surface energy and mass balance model"

_Geoscientific Model Development, 2020_

## Referee Comment (RC1) · Anonymous Referee #1 · 5 Mar 2020

COSIPY v1.2 - An open-source coupled snowpack and ice surface energy and mass balance model
Tobias Sauter, Anselm Arndt, and Christoph Schneider

This paper describes a distributed surface energy and mass balance model coded in Python and available as open source on github. The paper describes in much detail the physics included in the model.
The paper is well written and concise, although sometimes a bit too concise, see remarks below.

[Figure]

My main concern with this manuscript is that in my view it does not present anything new. There are several distributed energy and mass balance models available, some are more sophisticated than this one, some less, and at least one of them is also available as open source on github. The model itself is also not new, there are several publications with an earlier version of this model (Huintjes et al. 2014 and 2015), and the model physics in general is used in the other models as well and is already described in similar detail in other studies. I am not sure whether there are more of these type of models programmed in python, but that does not seem the key point here.

Thus, what makes this model special or new to warrant publication?

Besides this general concern, I have a few other comments.

General comments
It should be made much more clear what is new about this (see above).

In my experience it is often not so much the model formulation and running of it that is a problem, but the preparation of the input data. In this manuscript there is almost no information on how the input data is prepared and how it is distributed over the grid. Is this provided for in this package or should the user do that him/herself? And if it is included, how is it done? Make clear what the user is suposed to do him/herself and what is included.

Other information I am missing is on initial conditions, tuning and spin up. What procedure do you use? Is this also something provided for in the package or has the user do this him/herself?

[Figure]

After the model description, the model is applied to a Tibetan glacier as an example. I appreciate that you show that the model is indeed producing reasonable results, but I would have liked a bit more evaluation, analyses and interpretation on how well it is doing, and why there are differences, compared to observations and to other models.

Abstract
P1 Lines 1-7 are a very general introduction. Is that necessary in an abstract? I suggest to either remove it or make it much shorter. Formulations are also not clear. For example, 'key role' in what (line 1)? and where do 'these changes' (line 2) refer to?
P1 L8: remove 'lean'. I have no idea what you mean by this.
P1 L16: remove 'in'.

Introduction
P2 L2: What do you mean with 'many scientific aspects'?
P2: Note also the work by Ostby et al. 2017 TC, and by van Pelt et al. (several studies) for Svalbard.
P2 L30: The Hock and Holmgren 2005 JGl model is available on github.
P2 L33: Remove 'lean'.
P3 L6: Make much more clear what is new. I do not see it.

Model concept
eq(3): The second term on the richthand side reads: $k_s \frac{\delta^2 T_s}{\delta z^2}$ Shouldn't this be $\frac{\delta}{\delta z}\left(k_s \frac{\delta T_s}{\delta z}\right)$. In your case you ignore the effect of the gradient in k with depth. Furthermore, what is the functional form you take for $k_s$? And why use it? Bartelt and Lehning already note that they think this is an inferior description of $k_s$.
P3 L25: check the equation, for cs in combination with eq(3). I think there is a $\rho_s$ to many in eq(3).
P4 L7: Is your model indeed as deep that it reaches the base of the glacier? Most

models only go 20 to 30 m deep. More is not really necessary for climatic surface mass balance studies.

P4 L16: In my own experience, re-meshing, complete making of a new grid, is not necessary to do every time step, but can be made depended on melt and snow fall. This speeds up the model considerably when nothing is happening to the snowpack. Or do you also refer to re-meshing when only thickness of the layers changes a little, and thus also depth, due to densification?

P4 eq(5) Where does this equation come from? ColeÌ Ąou and Lesaffre, 1998, provides 1 equation for the full range of $\theta_i$

P5 L1: Does the model include saturation of the snow? And if so, how is it described, and if not, please mention.

P5 L2: What happens in the accumulation/firn area? When does runoff occur in that area?

P5 L17: Especially with respect to solar radiation it is important to mention how you distribute the input forcing over the glaciers. Do you include a formulation to distinguish between direct and diffuse radiation, shading? Or does the user have to do that separately?

P6 L2: Also in case of longwave radiation, how do you distribute this over the glacier? Do you then always use eq (16)?

P6 eq(17,18): I do not understand the term 1/Pr in this equation. In my opinion this factor should be included in how you calculate Ch and Ce, since not all methods that you present to calculate Ch and Ce should include this term.

P7 L1: How do you determine z0q and z0t, you only mention a factor. Do you indeed only apply a factor on z0m to obtain z0q or z0t or do you use a method such as described by Andreas, 1987, BLM?

P8 L4: chang to 10 superscrip (-2). Confusing as it is now.

P8 L11-15: What you describe here is a variation on what is described in Bartelt and Lehning. However, you refer here to a French report, which is hard to find. I prefer you to change the reference to something that is general available.

Example

P12 L10: Please make more clear that you start by running the model for a single location and only run it in distributed mode from line 30 onwards. It is often not clear whether you refer to a result for a single location or the whole glacier.

P12 L14: Capital RH instead of rH.

P12 L16: Where do you get the precipitation from?

P12 L24: You first have to mention that you obtain the surface temperature from Lout observations, else this statement makes no sense.

At this point I would like a bit more information on when the model is doing a good job, and when it struggles, and why. What are the limitations, how does this compare to other distributed mass and energy balance models?

P12 L24: Where is the stake you refer to here located with respect to the weather station and your grid point?

P12 L26: Typo: modelleld should be modelled.

P12 L24-30: In this analyses I suggest that you distinguish between the time the glacier is snow covered and when the ice surface appears. The model should be well capable to reproduce the amount of ice melt, whereas surface changes in case of snow cover are much more difficult, since that also includes firn densification processes. Presenting ice melt separately also gives an indication of how well you reproduce the energy fluxes at the surface. Unless this is all snow covered period. But you have to make that clear. Figure 1: Make clear whether this is m ice/snow or m w.e. And indicate in the figure when ice is exposed (or not).

P12 L33 - P13 L2: When you refer here to the distributed version of the model, do you compare the grid point of the weather station with the results when running the model only for the weather station location? If that is the case, why is there a difference in annual mass loss? What is done differently?

P13 L2: To what results do you refer here? Glacier wide? or Point location? Or the difference between them?

Conclusions

P15 L8: Remove 'of its kind'.

P15 L12-17: Other models can do this as well, and moste of the topics mentioned have been done, at least for individual regions. What does this model add to that?

―――――――――――――――――――

---

## Referee Comment (RC2) · Samuel Morin (Referee) · 14 Apr 2020

The manuscript by Sauter et al., describes the COSIPY v1.2 open-source coupled snowpack and ice surface energy and mass balance model. This model is designed to simulate the energy and mass balance of snow and ice covered surfaces, with applications for glacier mass balance simulations. The model builds on several decades of research in the field of snow cover and ice simulations. The main originality of this model is that it is implemented in Python. Given the scope and the content of the manuscript, it is fully appropriate for publication in Geoscientific Model Development. Overall, the manuscript reads well and I have not identified major flaws in the manuscript. Note,

however, that I haven't checked one by one all the equations in Sections 2 and 3, which are based on classical concepts and frameworks for snow and ice energy and mass balance. I have several comments, which can rather be seen as suggestions, to the authors, and a series of minor comments.

Main comments:

Section 4 : While Sections 2 and 3 are in fact of limited added-value given that the equations are concepts are already outlined in a number of previous publications (it is fine to leave them in the manuscript, this is a useful reference for users of the model or its output, perhaps complemented by recent publications such as Essery et al., 2013, http://dx.doi.org/10.1016/j.advwatres.2012.07.013 and Lafaysse et al., 2017, http://dx.doi.org/10.5194/tc-11-1173-2017), I find section 4, addressing "Model architecture", quite short and it could be expanded to better address the novelty and added-value of the model compared to previously existing models. For example, I think that it could be useful to provide more details regarding the Python libraries used for this model, their common dependencies, added-value, etc., and how the "modularity" of the model structure is implemented. This could be addressed not only by adding text, but also figures, providing an overview description of the model structure and the interlinkages between them.

Section 5 : The section 5 provides an example of the model use for the Zhadang glacier, High Mountain Asia, with illustrations of model output (Figures 1 and 2) and model performance (Figure 3) for this case study. The results appear to be reasonable for a typical energy and mass balance model applied to a glacier setting. However, this does not correspond to a full model evaluation exercise, and I think this model description article would greatly benefit from a more robust evaluation. In this respect, I think the dataset used for the ESM-SnowMIP intercomparison could be particularly useful. All the relevant data have been made available in Ménard et al., 2019 (https://doi.org/10.5194/essd-11-865-2019), and paper such as Krinner et al., 2018 (https://doi.org/10.5194/gmd-11-5027-2018) can be used as inspiration for providing

the evaluation indicators of snow models. Regardless of how this is handled, I consider useful for this model description article to provide some evaluation metrics relevant to the performance of the model described in this article.

Minor comments:

Page 4, line 15 : how do the re-meshing algorithms compare to existing re-meshing algorithms used in other snow cover models ? I think in particular of Crocus (Vionnet et al., 2012, https://doi.org/10.5194/gmd-5-773-2012), there are other models with re-meshing approaches. I think it would be good to position the approach taken here within other existing models.

Page 4, line 24 : "useful feature" : would it be possible to elaborate on what is meant by "useful feature" ? What metric was used to address the "usefulness" ?

Page 7, line 3 : I suggest to use the LaTeX symbol \varepsilon instead of \epsi, this seems to better match the graphical design of the "epsilon" symbol, when it refers to the emissivity.

Page 9, line 3, I suggest replacing "Von" by "von" for the name of "von Karman" (ideally with "accents" on the "a"s).

Page 10, line 7 : I don't think it is adequate to refer to "snow grain settling", but "Snow settling" would be less ambiguous and more accurate.

Page 11, line 20 : I think more details should be given on what is referred to here as "dynamic mesh" ?

Page 12, line 9 : More explanations could be given to better explain the content of the parenthesis "(not recommended for distributed simulations)"

Page 12, line 31 : I think more explanations are needed for "driven by ERA-5", in particular whether downscaling was applied, and if yes, how.

Page 13, Figure 1 : I suggest replacing "modelled" by "simulated" in the legend and

captions.

Page 14, Figure 3 : Would it be possible to provide a definition for the term "Speedup" ? I think this would be a useful clarification. If possible, it would be useful to provide a comparison of this metric with other existing models, in order to address to what extent the scalability of this Python-based model is comparable to implementations using other programming language.

Page 15, line 15 : I think it would be appropriate to also refer to multiphysics modelling, and it would be good to know to what extend COSIPY can be used for such applications (see e.g. Pritchard et al., 2020, https://doi.org/10.5194/tc-14-1225-2020).
* * *

---

## Author Comment (AC1) · 26 May 2020

We thank the reviewer for the constructive comments and suggestions. The comments are very helpful and will certainly strengthen the quality of the manuscript. Our response to the review can be found in the attached document.

**R**: Referee's comment
**A**: Author's response
**C**: Proposed changes in the manuscript
**Blue letters**: Suggested changes in the text
* * *
**General comments**

**R: This paper describes a distributed surface energy and mass balance model coded in Python and available as open source on github. The paper describes in much detail the physics included in the model. The paper is well written and concise, although sometimes a bit too concise, see remarks below.**

**My main concern with this manuscript is that in my view it does not present anything new. There are several distributed energy and mass balance models available, some are more sophisticated than this one, some less, and at least one of them is also available as open source on github. The model itself is also not new, there are several publications with an earlier version of this model (Huintjes et al. 2014 and 2015), and the model physics in general is used in the other models as well and is already described in similar detail in other studies. I am not sure whether there are more of these type of models programmed in python, but that does not seem the key point here. Thus, what makes this model special or new to warrant publication?**

**A:** Thank you for the thoughts. In fact, there are several distributed glacier mass balance models of varying complexity. The highest complexity is certainly reached by snow cover models (e.g. Snowpack, Crocus, etc.), some of which are freely accessible and actively maintained. COSIPY is a new edition of the obsolete Matlab version COSIMA (which was also developed by the first author). The differences between the models are, apart from the programming language, especially the model structure. This includes the discretization of the computational grid, the selection and implementation of the parameterizations, input/output routines, parallelization, etc. In addition, great emphasis was put on the documentation and readability of the code to offer other scientists the opportunity to actively participate in the further development. A documentation of this kind was not available in COSIMA. Since the differences between the versions are essential, and COSIPY already benefits from a relatively large community, we think that a citable article describing the model is needed. In our opinion, the GMD Journal is the appropriate platform for the description of new geoscientific models such as COSIPY.

In summary, here are the main points where COSIPY differs from other glacier mass balance models:

- is completely written in Python, modular and object-oriented
- completely based on open-source libraries
- has a readthedocs documentation (which is still in the development phase)
- parameterizations can be easily extended or modified by the user
- NetCDF IO
- easy integration of Weather Research and Forecast (WRF) forcing
- adapted for distributed glacier mass balances simulations; it needs to be pointed out in this regard that COSIPY is not a snow cover model

- has a community platform (Slack) and code is actively maintained
- new git commits are automatically tested via travis and codecov
- each model version gets a DOI
- has a restart option for operational applications

**R**: **It should be made much more clear what is new about this (see above).**

**A**: We have tried to highlight the differences (see comment above) between COSIPY and other models at several points in the text, such as:

P2L18-p3L6: "… *Ideally, a platform should (i) be continuously maintained, (ii) provide newly developed parameterisations, (iii) compile different model subversions developed for specific research needs, (iv) be easily extensible and (v) be well documented and readable. Here we present an open-source coupled snowpack and ice surface energy and mass balance model in Python (COSIPY) designed to meet these requirements. The structure is based on the predecessor model COSIMA (COupled Snowpack and Ice surface energy and MAss balance model, Huintjes et al., 2015). COSIPY provides a lean, flexible and user-friendly framework for modelling distributed snow and glacier mass changes. The framework consists of a computational core that forms the runtime environment and handles initialization, input-output (IO) routines, parallelization, and the grid and data structures. In most cases, the runtime environment does not require any changes by the user. Physical processes and parameterisations are handled separately by modules. The modules can be easily modified or extended to meet the needs of the end user. This structure provides maximum flexibility without worrying about internal numerical issues. The model is provided on a freely accessible git repository (https://github.com/cryotools/cosipy) and can be used for non-profit purposes. Scientists can actively participate in extending and improving the model code*".

We will expand this paragraph with additional information about the model structure and special features of COSIPY (see answer above).

**C**: (proposed changes to the original text are provided in **bold blue** letters) "… Ideally, a platform should (i) be continuously maintained, (ii) provide newly developed parameterisations, (iii) compile different model subversions developed for specific research needs, (iv) be easily extensible and (v) be well documented and readable. Here we present an open source coupled snowpack and ice surface energy and mass balance model in Python (COSIPY), which meets these requirements. The structure is based on the predecessor model COSIMA (COupled Snowpack and Ice surface energy and MAss balance model, Huintjes et al., 2015). COSIPY provides a lean, flexible and user-friendly framework for modelling distributed snow and glacier mass changes. The framework consists of a computing kernel that forms the runtime environment and handles initialization, input-output (IO) routines, parallelization, and grid and data structures. In most cases, the runtime environment does not require any changes by the user. **To increase the user friendliness, additional features are available to the user, such as a restart option for operational applications and automatic comparison between simulation and ablation stakes. The features will be further refined during the development phase.** Physical processes and parameterizations are handled separately by modules. The modules can easily be modified or extended to meet the needs of the end-user. This structure offers maximum flexibility without worrying about internal numerical issues. The model **is completely based on open-source libraries and** is provided on a freely accessible git repository (https://github.com/cryotools/cosipy) for non-profit purposes. Scientists can actively participate in extending and improving the model code. **Changes to the code are automatically tested with Travis CI (www.travis-ci.org) when uploaded to the repository. It is planned to publish updates**

**in regular intervals. To make working with COSIPY easier, a community platform (https://cosipy.slack.com) has been set up in addition to a detailed readthedocs documentation (https://cosipy.readthedocs.io/en/latest), allowing users and developers to exchange experiences, report bugs and communicate needs.**

**R: In my experience it is often not so much the model formulation and running of it that is a problem, but the preparation of the input data. In this manuscript there is almost no information on how the input data is prepared and how it is distributed over the grid. Is this provided for in this package or should the user do that him/herself? And if it is included, how is it done? Make clear what the user is suposed to do him/herself and what is included.**

**A**: We agree with the reviewer and identify the data pre-processing as one of the most important steps in the modelling process, but the pre-processing differs from case to case and the user's needs. For this reason, COSIPY does not provide any standard pre-processing routines and it is up to the user to prepare or spatially interpolate the data. However, we do provide example scripts that illustrate and facilitate the data preparation workflow for the user. An example is given in the online documentation (https://cosipy.readthedocs.io/en/latest/Documentation.html#quick-tutorial). The example shows how the input dataset can be generated from automatic weather station data and a given digital elevation model. The example scripts use simple lapse rates for temperature, precipitation and humidity for the interpolation. The wind speed, cloud cover and longwave radiation is assumed to be constant over the domain. For the interpolation of the radiation the radiation model of Wohlfahrt et al (2016) is used (doi: 10.1016/j.agrformet.2016.05.0120).

**C**: Chapter 4.1 deals explicitly with the input/output and refers to the corresponding website. In this chapter it says:

"*The model is driven by meteorological data that must be provided in a corresponding NetCDF file (see https://cosipy.readthedocs.io/en/latest/Ressources.html). Input parameters include atmospheric pressure, air temperature, cloud cover fraction, relative humidity, incoming shortwave radiation, total precipitation and wind velocity. Optional snowfall and incoming longwave radiation can be used as forcing parameters. In addition to meteorological parameters, COSIPY requires static information such as topographic parameters and a glacier mask. **Example workflows for creating and converting static and meteorological data into the required NetCDF input is included in the source code (https://cosipy.readthedocs.io/en/latest/Documentation.html#quick-tutorial)**. Besides the standard output variables, there is also the possibility to store vertical snow profile information (not recommended for distributed simulations). To reduce the amount of data, the users can specify which of the output variables will be stored*.*"

We will change the phrase 'Various tools are available ...' to 'Example workflows for creating and converting static and meteorological data into the required NetCDF input is included in the source code (https://cosipy.readthedocs.io/en/latest/Documentation.html#quick-tutorial)'.

**R: Other information I am missing is on initial conditions, tuning and spin up. What procedure do you use? Is this also something provided for in the package or has the user do this him/herself?**

**A**: Simulations depend on the initial conditions and the spin-up time. To ensure maximum flexibility, these must be specified by the user by choosing an adequate simulation period and initial conditions. As with all models, the model is calibrated by adjusting the model parameters and constants. The user has the possibility to adjust all parameters and constants of the parameterization in the configuration file. Which metrics users want to use for the evaluation depends on their specific application. By default, COSIPY automatically calculates the root mean squared error between the simulation and ablation measurements, if the measurements are provided.

**C**: We will rename the title of Section 4.1 to '**IO and initial condition**' and add the following sentences: '... can be used as forcing parameters. **If the snow height (or snow water equivalent) and/or surface temperature are also specified in the input file, these are used as initial conditions. Otherwise, snow depth and surface temperature are assumed to be homogenous in space at the start of the simulation according to the specifications in the configuration file**.'

**R**: **After the model description, the model is applied to a Tibetan glacier as an example. I appreciate that you show that the model is indeed producing reasonable results, but I would have liked a bit more evaluation, analyses and interpretation on how well it is doing, and why there are differences, compared to observations and to other models.**

**A**: In this contribution we focus on the model description, but of course evaluation and interpretation are very important as well. It is difficult to find good glaciological data to compare different model versions but we are going to add the comparison of ablation in specific periods between COSIPY output and ablation stake readings. Furthermore, we will include profile plots of snow layer properties using data from Hintereisferner in the European Alps as a second example. We will also consider using data of the ESM-SnowMIP intercomparison project as suggested by the other reviewer. However, we emphasize that COSIPY is a glacier mass balance model and not a snow model so that a specific difficulty could be the prescribed soil temperature in the ESM-SnowMIP project. Nevertheless, we may use some of the metrics for the evaluation of COSIPY. However, a model intercomparison clearly is beyond the scope of this contribution.

**C**: We will make the appropriate changes to the existing text in Chapter 5 and will introduce new paragraphs/sub-chapters on the new datasets and evaluations.
* * *
**Abstract**

**R**: **P1 Lines 1-7 are a very general introduction. Is that necessary in an abstract? I suggest to either remove it or make it much shorter. Formulations are also not clear. For example, 'key role' in what (line 1)? and where do 'these changes' (line 2) refer to?**

**A**: The comments of the expert are reasonable. Since GMD is not a glaciological journal, we thought to create a broader context why distributed mass balance models are needed. But if this context is too broad we will shorten the first lines.

**C**: "**Glacier changes are a vivid example of how environmental systems react to a changing climate.** Distributed surface mass balance models which translate the meteorological conditions on glaciers into local melting rates **help to** attribute and detect glacier mass and volume responses to changes in the **climate drivers**. A well ...."

**R**: **P1 L8: remove 'lean'. I have no idea what you mean by this.**

**A**: The term "lean" is derived here from "lean concept".  A lean design understands the requirements of the model user and focuses on continuously improving the handling of the model without unnecessarily expanding the model environment.

**C**: If this term is confusing we will remove it.

**R**: **P1 L16: remove 'in'.**

**A/C**: Will be done.
* * *
**Introduction**

**R**: **P2 L2: What do you mean with 'many scientific aspects'?**

**A**: The chosen expression is probably unfortunate. What was meant was rather the perspectives on various scientific questions.

**C**: We will replace 'scientific aspects' with the term '**scientific issues**'.

**R: P2: Note also the work by Ostby et al. 2017 TC, and by van Pelt et al. (several studies) for Svalbard.**

**A**: We are aware that there are very good and mentionable studies on this topic. We have tried to present a selection that covers the wide range of mass balance studies. If one or the other study is not listed, it is not intentional.

**A/C**: We will add the work of Ostby et al. 2017 and van Pelt to the reference list.

**R: P2 L30: The Hock and Holmgren 2005 JGI model is available on github.**

**A/C**: We are aware that this model is available on github and we have mentioned this work in line 28.

**R: P2 L33: Remove 'lean'.**

**A/C**: If this term is confusing we will remove it (see comment above).

**R: P3 L6: Make much more clear what is new. I do not see it.**

**A/C**: See response above in general comments.
* * *
**Model concept**

**R**: **eq(3): The second term on the right hand side reads:** $k_s \frac{\delta^2 T_s}{\delta z^2}$ **. Shouldn't this be** $\frac{\delta}{\delta z}(k_s \frac{\delta T_s}{\delta z})$. **? In your case you ignore the effect of the gradient of k with depth. Furthermore, what is the functional form you take for** $k_s$ **? And why use it? Bartelt and Lehning already note that they think this is an inferior description of** $k_s$ **.**

**A**: In general form this equation should indeed reads as $\frac{\delta}{\delta z}(k_s \frac{\delta T_s}{\delta z})$ but since in our model $k_s$ does not depend on T, $k_s$ is assumed to be a constant (average over the considered layers where the derivative is calculated - hence its a bulk conductivity). Thus the equation reduces to $k_s \frac{\delta^2 T_s}{\delta z^2}$. This simplifies the calculation and allows for solving the equation using a linear equation system. We agree that better results may be obtained when $k_s$ depends on the spatial variable, e.g. $k_s(z)$. The equation becomes nonlinear and slightly more complicated. A gauge transformation could eliminate the spatial dependency and reduce the equation to $k_s \frac{\delta^2 T_s}{\delta z^2}$. Right now this is not implemented in the model, so that the given equation is correct. However, we agree that we should keep in mind that using a nonlinear heat equation would be an improvement of the model.

We are not exactly sure what the reviewer means with functional form and 'why use it'. The comment probably relates to the calculation of $k_s$. As given on p3L25, the volumetric thermal conductivity is calculated by the volumetric fractions of ice, water and air. The thermal conductivities for the constitutes are assumed to be constant (values are given in Appendix A).

Bartelt and Lehning indeed find the volumetric conductivity inferior to empirical or microstructural thermal conductivity models. The latter cannot be implemented as COSIPY does not model the microstructure of snow or ice. An option would be an empirical form of $k_s$ depending on density and/or temperature. For sake of consistency with COSIMA, we will add the empirical form, $k_s = 0.021 + 2.5(\rho_s/1000)^2$, suggested by Anderson (1976). The user can then choose between the two forms.

**C**: We will point out in p3L25 that this is a bulk thermal conductivity, given by the average of the involved layers, and that a linear system of equations is used to solve the heat equation. We will also add the empirical thermal conductivity equation based on density (see above) with the note that the user can choose between these two options.

**R**: **P3 L25: check the equation, for cs in combination with eq(3). I think there is a** $\rho_s$ **to many in eq(3).**

**A**: Thank you for pointing this out.

**C**: We have removed the fractional densities from $c_s$.

**R**: **P4 L7: Is your model indeed as deep that it reaches the base of the glacier? Most models only go 20 to 30 m deep. More is not really necessary for climatic surface mass balance studies.**

**A**: In fact, the domain does not always go to the base of the glacier and the statement is wrong. The user can determine the maximum depth of the computing domain. The default setting is 20 m.

**C**: We'll rewrite the sentence to '***At the bottom of the domain***, …'.

**R**: **P4 L16: In my own experience, re-meshing, complete making of a new grid, is not necessary to do every time step, but can be made depended on melt and snow fall. This speeds up the model considerably when nothing is happening to the snowpack. Or do you also refer to re-meshing when only thickness of the layers changes a little, and thus also depth, due to densification?**

**A**: If the logarithmic approach is chosen, the remeshing is executed at each time step. This means that every change in layer thickness due to settling, densification, snowfall or melt triggers the remeshing algorithm. The logarithmic profile is defined by the thickness of the top layer and a stretching which is specified by the user. So far we have not thought about making the logarithmic remeshing conditional. But we will try this and hope for a speedup. Thanks for the helpful hint.

**C**: Within the scope of this COSIPY version we refrain from a conditional remeshing algorithm, but will test it in the next version. Therefore we see no reason to change the manuscript at this point.

**R**: **P4 eq(5) Where does this equation come from? Coleou and Lesaffre, 1998, provides 1 equation for the full range of**

**A**: This is the same equation used by the study of Wever et al. (2014), which used exactly this formulation.

**C**: We will add this reference.

**R**: **P5 L1: Does the model include saturation of the snow? And if so, how is it described, and if not, please mention.**

**A/C**: Yes. But maybe we don't really understand the question. Each layer can retain water up to its retention capacity (see Eq. 5). Only if the capacity is exceeded, the excess water is transported to the next layer. This approach corresponds to the commonly used bucket approach. When the liquid water content reaches the retention capacity, the snow is saturated. How this is treated in the model is described in the text (p4L27): "In case the liquid water content of a layer exceeds its retention capacity … the excess water is drained into the subsequent layer (bucket approach)".

**R**: **P5 L2: What happens in the accumulation/firn area? When does runoff occur in that area?**

**A**: This is an exciting question. Up to now, a snow-ice threshold value can be defined by the user, which determines from which density on snow is referred to as ice. The threshold is usually set around 900 kg m$^{-3}$. Water percolates up to the first layer that is greater than or equal to this density and is then regarded as runoff. If there is no such layer, water percolates through the lower boundary of the domain and is then considered as runoff. We will clarify in the text how water percolation is treated in the accumulation/firn area.

**C**: "The liquid water is passed on until it reaches **either a layer of ice or** the surface of the glacier, where it is considered to be runoff. **For this purpose a threshold value was introduced which defines the transition from snow to ice. If no such layer exists, water is passed on until it reaches the lower limit of the domain and is then considered as runoff**."

**R: P5 L17: Especially with respect to solar radiation it is important to mention how you distribute the input forcing over the glaciers. Do you include a formulation to distinguish between direct and diffuse radiation, shading? Or does the user have to do that separately?**

**A**: We agree. As mentioned above, COSIPY does not provide any standard pre-processing routines and it is up to the user how to prepare or spatially interpolate the data. However, we do provide example scripts that illustrate and facilitate the data preparation workflow for the user. An example is given in the online documentation. The example shows how the input dataset can be generated from automatic weather station data and a given digital elevation model. The example scripts use simple lapse rates for temperature, precipitation and humidity for the interpolation. The wind speed is assumed to be constant over the domain. For the interpolation of the radiation the radiation model of Wohlfahrt et al (2016) is used (doi: 10.1016/j.agrformet.2016.05.0120). The current implementation does not distinguish between direct and diffuse radiation, but considers the total incoming solar radiation.

**C**: We will add a sentence in chapter 4 to clarify that the pre-processing of the data must be done by the user: The paragraph now reads as:

"*The model is driven by meteorological data that must be provided in a corresponding NetCDF file (see https://cosipy.readthedocs.io/en/latest/Ressources.html). Input parameters include atmospheric pressure, air temperature, cloud cover fraction, relative humidity, incoming shortwave radiation, total precipitation and wind velocity. Optional snowfall and incoming longwave radiation can be used as forcing parameters. In addition to meteorological parameters, COSIPY requires static information such as topographic parameters and a glacier mask. Example workflows for creating and converting static and meteorological data into the required NetCDF input is included in the source code (https://cosipy.readthedocs.io/en/latest/Documentation.html#quick-tutorial). Besides the standard output variables, there is also the possibility to store vertical snow profile information (not recommended for distributed simulations). To reduce the amount of data, the users can specify which of the output variables will be stored.*"

We will change the phrase 'Various tools are available ...' to '*Example workflows for creating and converting static and meteorological data into the required NetCDF input is included in the source code (https://cosipy.readthedocs.io/en/latest/Documentation.html#quick-tutorial)*'.

**R**: **P6 L2: Also in case of longwave radiation, how do you distribute this over the glacier? Do you then always use eq (16)?**

**A**: There are two possibilities here. The user can specify the distributed longwave radiation in the input data or the longwave radiation is calculated using the Stephan-Boltzmann law and atmospheric emissivity (Eq. 15). Eq. 16 is included in the calculation of the atmospheric emissivity. As with all other input data, the longwave radiation must be distributed over the topography by the user.

**C:** See comment above.

**R**: **P6 eq(17,18): I do not understand the term 1/Pr in this equation. In my opinion this factor should be included in how you calculate Ch and Ce, since not all methods that you present to calculate Ch and Ce should include this term.**

**A**: Thank you very much for this hint. This is actually an error in the formulation. We will remove 1/Pr from the Eq. (17) and Eq. (18) and also correct the Equations 25-27.

**C**: We will correct the Equations 17, 18 and 25-27.

**R**: **P7 L1: How do you determine z0q and z0t, you only mention a factor. Do you indeed only apply a factor on z0m to obtain z0q or z0t or do you use a method such as described by Andreas, 1987, BLM?**

**A**: The two roughness lengths z0q and z0t are derived from z0v and are in fact one or two orders of magnitude smaller. Currently, the two roughness lengths are not parameterized separately as indicated in the text: "*The aerodynamic roughness length z0v is simply a function of time and increases linearly for snowpacks from fresh snow to firn (Mölg et al., 2012). For glaciers, z0v is set to a constant value. According to the renewal theory for turbulent flow, z0q and z0t are assumed to be one and two orders of magnitude smaller than z0v, respectively (Smeets and van den Broeke, 2008; Conway and Cullen, 2013)*".

**C**: We think these sentences adequately describe how the roughness lengths are derived and thus do not make any changes.

**R**: **P8 L4: chang to 10 superscrip (-2). Confusing as it is now.**

**A/C**: Will be done.

**R: P8 L11-15: What you describe here is a variation on what is described in Bartelt and Lehning. However, you refer here to a French report, which is hard to find. I prefer you to change the reference to something that is general available.**

**A**: As far as we know, Bartelt and Lehning use a microstructure based viscosity formulation. Eq. 32 originates from Anderson (1976) but the parameter values used by default are those from Boone (2004). One way around this reference would be to cite another paper that used the same parameterization, e.g. Essery et al (2013).

**C**: We will, therefore, also include the reference Essery et al. (2013) in this paragraph.
* * *
**Example**

**R**: **P12 L10: Please make more clear that you start by running the model for a single location and only run it in distributed mode from line 30 onwards. It is often not clear whether you refer to a result for a single location or the whole glacier.**

**A**: Yes we agree that this has to be better pointed out in the text.

**C**: We will change P12 L12 to: *"**As a first example, we** use hourly data from May 2009 to June 2012 from an automatic weather station (AWS) on the Zhadhang Glacier (Huintjes et al.,2015) **fo force COSIPY as a point model for a single location**."* and P12 L30 to: *"**For a distributed glacier-wide run we drive COSIPY by ERA5 data instead of in-situ observations.** The glacier-wide cumulative surface mass balance for the decade 2009 to 2018 is presented in Figure 2. The computational domain consisted of 1837 grid cells with a spatial resolution of approximately 30 m (1 arcsecond) (see Fig. 2).*

**R**: **P12 L14: Capital RH instead of rH.**

**A/C**: We will change the abbreviation for relative humidity to capital RH.

**R**: **P12 L16: Where do you get the precipitation from?**

**A**: We got the accumulated precipitation from a sonic ranger.

**C**: We will change the sentence to: *"The relevant variables air pressure $p_{zt}$ , air temperature $T_{zt}$ , relative humidity $rH_{zt}$ , incident short-wave radiation $qG$, **snowfall SF** and wind speed $u_{zv}$ were measured by the AWS."*

**R**: **P12 L24: You first have to mention that you obtain the surface temperature from Lout observations, else this statement makes no sense. At this point I would like a bit more information on when the model is doing a good job, and when it struggles, and why. What are the limitations, how does this compare to other distributed mass and energy balance models?**

**A**: Yes that is true. We will mention that the surface temperature is obtained from longwave radiation measurements. Beyond that we cannot compare in detail to other models without running these on the same dataset. However, a model intercomparison is beyond the scope of this manuscript.

**C**: We will change the sentence (P12 L21) to: *"Figure 1a and 1b show the glacier surface temperatures **determined from longwave radiation measurements** for two periods where in-situ measurements were available."*
As mentioned at the beginning of the document for the general comment on the Zhadang example, we will make the appropriate changes to the existing text in Chapter 5 and will introduce new paragraphs/sub-chapters on the new datasets and evaluations.

**R**: **P12 L24: Where is the stake you refer to here located with respect to the weather station and your grid point?**

**A**: The stake is in the vicinity of the weather station. It is located  within the same grid point of the simulation as the weather station. We will add more stake data to the evaluation of the results and point out much clearer where the stakes are located in the revised version.

**C**: According to the answer above, we will change the whole paragraph concerning the ablation stakes and compare only melt periods between model and observations.

**R**: **P12 L26: Typo: modelleld should be modelled.**

**A/C**: Will be changed to modelled.

**R**: **P12 L24-30: In this analyses I suggest that you distinguish between the time the glacier is snow covered and when the ice surface appears. The model should be well capable to reproduce the amount of ice melt, whereas surface changes in case of snow cover are much more difficult, since that also includes firn densification processes. Presenting ice melt separately also gives an indication of how well you reproduce the energy fluxes at the surface. Unless this is all snow covered period. But you have to make that clear. Figure 1: Make clear whether this is m ice/snow or m w.e. And indicate in the figure when ice is exposed (or not).**

A: In Figure 1, it is m ice/snow not m w.e. We will clarify this in the capiton and we agree that this will be a valuable improvement to the manuscript to better distinguish between snow melt and ice melt. As mentioned above, we will add the comparison between single melt periods which we can constrain using stake readings and compare the ablation phase between the model and the stake readings for those.

C: According to the answer above, we will make the appropriate changes to the existing text in Chapter 5 and will introduce new paragraphs/sub-chapters on the new datasets and evaluations.

**R**: **P12 L33 - P13 L2: When you refer here to the distributed version of the model, do you compare the grid point of the weather station with the results when running the model only for the weather station location? If that is the case, why is there a difference in annual mass loss? What is done differently?**

**A**: Thank you for pointing this out. We will change this in the text because it does not make sense to compare the distributed glacier-wide mass balance to the single point simulation at the weather station.

**C**: We will compare here the glacier-wide run only with the literature values and not with the run for the location of the weather station. We will skip the comparison with the weather station location and change P12 L32 - P13 L3 to::
*"The simulated mass balance during this period was −1.5 m w.e. a⁻¹. The results are in line with the analysis of Qu et al. (2014) who reported negative mass balances of −1.9, −2.0, −0.8 and −2.7 m w.e for the years 2009 to 2012."*

**R**: **P13 L2: To what results do you refer here? Glacier wide? or Point location? Or the difference between them?**

**A**: In this case we refer to the glacier wide run. We deleted the sentence with the comparison to the weather station data. After that correction it is more obvious that from P12 L30 on we talk only about the distributed ERA5-forced run.

**C**: See proposed changes to reviewer comment above.
* * *
**Conclusion**

**R**: **P15 L8: Remove 'of its kind'.**

**A/C**: Will be done.

**R**: **P15 L12-17: Other models can do this as well, and moste of the topics mentioned have been done, at least for individual regions. What does this model add to that?**

**A**: There are a large number of mass balance models for snow and glaciers, but as mentioned at the beginning of this revision, there are significant differences with regard to implementation and model structure. COSIPY does not compete with existing models, but offers an accessible model structure with commonly used parameterization in glaciology, while micro-structural process required for detailed snow simulation are neglected (e.g. the micro-structure) as they are built into more sophisticated models such as SNOWPACK or CROCUS. The implementation in Python and the easy access for users makes the model attractive for glaciological applications which is also reflected in the increasing number of users.

**C**: We will summarize these advantages as well as the disadvantages in the conclusion.

---

## Author Comment (AC2) · 26 May 2020

We thank the reviewer for the constructive comments and suggestions. The comments are very helpful and will certainly strengthen the quality of the manuscript. Our response to the review can be found in the attached document.

**R**: Referee's comment
**A**: Author's response
**C**: Proposed changes in the manuscript
**Blue letters**: Suggested changes in the text
* * *
**R: The manuscript by Sauter et al., describes the COSIPY v1.2 open-source coupled snowpack and ice surface energy and mass balance model. This model is designed to simulate the energy and mass balance of snow and ice covered surfaces, with applications for glacier mass balance simulations. The model builds on several decades of research in the field of snow cover and ice simulations. The main originality of this model is that it is implemented in Python. Given the scope and the content of the manuscript, it is fully appropriate for publication in Geoscientific Model Development. Overall, the manuscript reads well and I have not identified major flaws in the manuscript. Note, however, that I haven't checked one by one all the equations in Sections 2 and 3, which are based on classical concepts and frameworks for snow and ice energy and mass balance. I have several comments, which can rather be seen as suggestions, to the authors, and a series of minor comments.**

**Section 4 : While Sections 2 and 3 are in fact of limited added-value given that the equations are concepts are already outlined in a number of previous publications (it is fine to leave them in the manuscript, this is a useful reference for users of the model or its output, perhaps complemented by recent publications such as Essery et al., 2013, http://dx.doi.org/10.1016/j.advwatres.2012.07.013 and Lafaysse et al., 2017, http://dx.doi.org/10.5194/tc-11-1173-2017), I find section 4, addressing "Model architecture", quite short and it could be expanded to better address the novelty and added value of the model compared to previously existing models. For example, I think that it could be useful to provide more details regarding the Python libraries used for this model, their common dependencies, added-value, etc., and how the "modularity" of the model structure is implemented. This could be addressed not only by adding text, but also figures, providing an overview description of the model structure and the interlinkages between them.**

**A**: We will complement the Sections with the work done by Essery et al. (2013) and Lafaysse et al. (2017) at the corresponding locations. Both are excellent references.

Originally we had planned another paragraph about the modular structure of the model. However, we decided to not integrate it, as we felt that this technical information would be better presented in the readthedocs online documentation (https://cosipy.readthedocs.io/en/latest). We will, however, add a paragraph about the code implementation, dependencies and structure of the model. Some information is already given in Section 7 'Code availability, documentation, and software requirements', which is mandatory in GMD. Additionally, we will also discuss the added-value of COSIPY compared to previous glacier mass balance models. This has already been criticized by the first reviewer.

**C**: In Section 4, we will take up this issue again and highlight the special features. These include

- is completely written in Python, modular and object-oriented
- completely based on open-source libraries

- has a readthedocs documentation (which is still in the development phase)
- parameterizations can be easily extended or modified by the user
- NetCDF IO routines
- easy integration of Weather Research and Forecast (WRF) forcing
- adapted for distributed glacier mass balances simulations; it needs to be pointed out in this regard that COSIPY is not a snow cover model
- has a community platform (Slack) and code is actively maintained
- new git commits are automatically tested via travis and codecov
- each model version gets a DOI
- has a restart option for operational applications

**R: Section 5 : The section 5 provides an example of the model use for the Zhadang glacier, High Mountain Asia, with illustrations of model output (Figures 1 and 2) and model performance (Figure 3) for this case study. The results appear to be reasonable for a typical energy and mass balance model applied to a glacier setting. However, this does not correspond to a full model evaluation exercise, and I think this model description article would greatly benefit from a more robust evaluation. In this respect, I think the dataset used for the ESM-SnowMIP intercomparison could be particularly useful. All the relevant data have been made available in Ménard et al., 2019 (https://doi.org/10.5194/essd-11-865-2019), and paper such as Krinner et al., 2018 (https://doi.org/10.5194/gmd-11-5027-2018) can be used as inspiration for providing the evaluation indicators of snow models. Regardless of how this is handled, I consider useful for this model description article to provide some evaluation metrics relevant to the performance of the model described in this article.**

**A**: Thank you for the suggestion. We agree it would be of great benefit to include a comparison with other models, however a full model intercomparison is beyond the scope of this manuscript.. We will use the data from Ménard et al., 2019 as forcing data to reproduce the metrics and compare COSIPY to other models similar to Krinner et al., 2018. It might be that we can not reproduce all metrics since COSIPY is a glacier energy and mass balance model and not a snowmodel, i.e., the soil heat flux is not parameterized in the same way, for example. Furthermore, we will extend the model evaluation for the Zhadang glacier with more ablation-stake data and present profile plots of the layer properties for the Hintereisferner in the European Alps. This exercise is also in response to comments of the other reviewer.

**C**: We will make the appropriate changes to the existing text in Chapter 5 and will introduce new paragraphs/sub-chapters on the new datasets and evaluations.

**R: Page 4, line 15 : how do the re-meshing algorithms compare to existing re-meshing algorithms used in other snow cover models ? I think in particular of Crocus (Vionnet et al., 2012, https://doi.org/10.5194/gmd-5-773-2012), there are other models with remeshing approaches. I think it would be good to position the approach taken here within other existing models.**

**A:** Similar to CROCUS, COSIPY uses a set of criteria that determine when two layers are merged or splitted. As already indicated in the text the user can choose between two options:

(i) Logarithmic profile: This method is only suitable for simulations where the layering of the snowpack is not relevant (bulk). Layer thicknesses are calculated starting from the top layer, which always remains at a constant thickness, and gradually increases with depth by a constant stretching factor. Thus the layers close to the surface have a higher spatial resolution, which is advantageous for the computation of the energy and mass fluxes at the surface.

(ii) Adaptive profile: The adaptive algorithm runs in three consecutive steps: (1) adding/removing snow/ice at the surface, (2) adjusting the first layer, (3) updating internal layers.

In the first step it is checked whether snow falls or melts away (note: internal layers can also melt). If snow falls on the glacier surface, it will only remain on the surface if it reaches a user-defined minimum snow thickness. If it falls on an existing snowpack, any snowfall that exceeds a user-defined minimum threshold is added to the snowpack. Melt is removed from the first layers and internal layers. After this step, layers can become very small and the thickness of the first layer no longer corresponds to the user-specified constant thickness. Therefore, it is necessary to remesh the layers.

In the second step, the top layer is adjusted first. The top layer is remeshed so that this layer always has the user-defined layer thickness (default value is 0.01 m). The adaptation of the top layers together with internal melting processes can reduce the internal layers to a very low thickness. To avoid thin layers, the layers are merged or split in the last step (see next paragraph).

In the last step, internal layers are splitted or merged. For each layer, a check is made to identify layers with a thickness of less than a defined minimum layer thickness. Such thin layers are merged with the layer below. Also if the differences in temperature and density of two subsequent layers are less than a user defined threshold (similarity criteria), they will be merged. How often a merging/splitting can take place per time step is also defined by the user (correction steps). Unlike CROCUS, internal remeshing always starts from the surface, i.e. the uppermost layers are adapted first. Depending on how many correction steps are set by the user, it can happen that only the uppermost layers are remeshed.

**C:** We will extend the description of the meshing algorithms (beginning from page 4 line 15) and describe them in more detail as outlined above.

**R: Page 4, line 24 : "useful feature" : would it be possible to elaborate on what is meant by "useful feature" ? What metric was used to address the "usefulness" ?**

**A:** The term 'useful feature' should indicate that the adaptive algorithm is reasonable when one is interested in the stratification of the snowpack. In contrast to the logarithmic profile, one obtains well resolved layers. This can be important for some glaciological issues, but it is computationally more expensive than the logarithmic algorithm.

**C:** We will rewrite the sentence "*The adaptive re-meshing proves to be a useful feature, but slightly increases both the computing time and the data volume*" to "***Unlike the logarithmic approach, adaptive re-meshing resolves individual layers but slightly increases both computing time and data volume.***"

**R: Page 7, line 3 : I suggest to use the LaTeX symbol \varepsilon instead of \epsi, this seems to better match the graphical design of the "epsilon" symbol, when it refers to the emissivity.**

**A/C:** Will be done.

**R: Page 9, line 3, I suggest replacing "Von" by "von" for the name of "von Karman" (ideally with "accents" on the "a"s).**

**A/C:** Will be done.

**R: Page 10, line 7 : I don't think it is adequate to refer to "snow grain settling", but "Snow settling" would be less ambiguous and more accurate.**

**A/C:** We agree and will change it to "snow settling".

**R: Page 11, line 20 : I think more details should be given on what is referred to here as "dynamic mesh" ?**

**A:** The wording is probably a bit confusing. The term should refer to the computational mesh which can be (dynamically) adapted by the re-meshing algorithms.

**C:** To clarify this misunderstanding we will remove the term 'dynamic'.

**R2: P12 L9: More explanations could be given to better explain the content of the parenthesis "(not recommended for distributed simulations)"**

**A**: The user can specify in a file which data should be stored. In addition to the atmospheric variables, COSIPY calculates the state of the snow/ice layers. Since the number of vertical layers and grid cells can be very high, it is recommended to store only those variables that are necessary for later evaluation. We recommend dropping the states of the layers in distributed simulations to save memory space.

**C**: We will change the sentence to: "*Besides the standard output variables there is also the possibility to store vertical snow profile information, although to save memory we can only recommend this for single point simulations.*"

**R2: P12 L31 : I think more explanations are needed for "driven by ERA-5", in particular whether downscaling was applied, and if yes, how.**

**A**: We extracted the needed input data from the nearest ERA5 grid point and applied downscaling methods to the variables. We will further clarify this in the revised version of the manuscript.

**C**: We will add a table to the manuscript with the applied downscaling approaches and change the sentence to: *"The model was driven by ERA5 data instead of in-situ observations.*

*The ERA5 data were downscaled to the site using straightforward approaches. Temperature, $T_{zr}$, and humidity, $RH_{zt}$, were corrected to the altitude of the grid cell using empirical lapse rates. For pressure, $p_{zt}$, the barometric formula was used. The radiation model of Wohlfahrt et al (2016) was used for the incoming shortwave radiation to account for effects of shadowing, slope and aspect. Total precipitation RRR, N and $U_{zv}$ were used directly from the closest ERA5 grid point."*

**R2: P13  Figure 1: I suggest replacing "modelled" by "simulated" in the legend and captions.**

**A/C**: We agree and will change it accordingly in all legends and captions.

**R2: Page 14, Figure 3: Would it be possible to provide a definition for the term "Speedup"? I think this would be a useful clarification. If possible, it would be useful to provide a comparison of this metric with other existing models, in order to address to what extent the scalability of this Python-based model is comparable to implementations using other programming language.**

**A**: Thank you for the important comment. The speedup is the ratio between the single-core execution time and the execution time of the corresponding multiple-core simulation. We wanted to point this out with the sentence: "..., i.e. the ratio of the original execution time with the execution time of the corresponding node test." In the present case, it is difficult to compare this value with other models because we would have to run the other models to the same test case. The speedup in the present case should rather show the performance gain when using multiple cores in contrast to a single core computer setup.

**C**: We will change the caption of Figure 3 to: *"Speedup (execution time of single-core simulation divided by execution time of the corresponding multiple-core simulation) for computing a 10-year distributed COSIPY run on Zhadang glacier with 206 grid points."* and the respective sentence to: *"..., i.e. the ratio of the original execution time (single core) with the execution time of the corresponding test (multiple cores)."*

**R: Page 15, line 15 : I think it would be appropriate to also refer to multiphysics modelling, and it would be good to know to what extend COSIPY can be used for such applications (see e.g. Pritchard et al., 2020, https://doi.org/10.5194/tc-14-1225-2020).]**

**A:** Multiphysical modeling is a very exciting topic and we are convinced that it will become even more important in the future. COSIPY is a modeling platform designed to test and apply different parameterizations. In principle it is already possible to generate ensemble simulations with different physical parameterizations and solvers, but COSIPY is not yet an ensemble multiphysics modeling environment. As a vision for the future it is conceivable to extend COSIPY for automatic ensemble simulations. Various uncertainties could be included - multiphysics modeling, perturbed input data or parameter uncertainty. So far, we have not yet thought about how an ensemble can be realized in a

single simulation. Until we have a feasible idea, we have no choice but to run COSIPY with different combinations of physical parameterizations or input uncertainty and evaluate the statistics afterwards.

**C:** At this point we will, as an addition to the existing manuscript, only provide an outlook on what might be possible with COSIPY in the future, e.g. ensemble simulations.

---

## Author Response (AR2)

**Author response**

R: Referee's comment
A: Author's response
C: Proposed changes in the manuscript
S: Status
Blue letters: Suggested changes in the text
* * *
**R: P4 L27: Add ($\theta _e$) behind 'retention capacity' in order to introduce the parameter. Also add that it represents the fraction of the fraction pore space filled with water.**

A: We replaced 'retention capacity' by irreducible water content. The variable name is added here as an insertion between two commas. It is therefore no longer necessary to add the symbol after the designation.

**R: P4: Regarding saturation, I wondered whether in this model it is possible to have saturation with respect to the total pore space, i.e. $\theta _e = 1$? Thus if a slush layer is allowed to develop.**

A: No, the model does not allow theta_w to be greater than thete_e, hence no slush layer can develop.

**R: P17 L4: Better to remove 'in the next months' and replace it with something like 'in the autumn of 2020'.**

A: We changed the text to: ''The system is currently running in test mode but will be available to the public in spring 2021.''

**R: P18-19: In the conclusions, I would still like it when it is made more clear that COSIPY does not include the distribution of input over a grid, nor does it do any input preparation.**

A: Yes, we have changed the text to: ''The model is written in Python and completely based on open source libraries. The model, source code, case studies and codes examples for data preprocessing are provided on a freely accessible Git repository (\url{https://github.com/cryotools/cosipy}) for non-profit purposes.''